



**Energetic electron enhancements under radiation belt (L < 1.2) during**
**nonstorm interval on August 1, 2008**
Alla V. Suvorova [1,3], Alexei V. Dmitriev[2,3], and Vladimir A. Parkhomov[4]
[1] GPS Science and Application Research Center, National Central University, Jhongli, Taiwan
[2] Institute of Space Science, National Central University, Jhongli, Taiwan
[3] Skobeltsyn Institute of Nuclear Physics, Lomonosov Moscow State University, Moscow,
Russia
[4] Baikal State University, Irkutsk, Russia
*Correspondence to*: Alla Suvorova (suvorova_alla@yahoo.com)
**Abstract**
An unusual event of deep injections of >30 keV electrons from the radiation belt to low L shells
(L <1.2) in midnight-dawn sector occurred during nonstorm conditions on August 1, 2008. Using
THEMIS observations in front of the bow shock, we found transient foreshock conditions and
rotational discontinuities passing the subsolar region at that time. These conditions resulted in
generation of fast magnetosheath plasma jets and penetration of the magnetosheath plasma into
the magnetosphere as were observed by the THEMIS probes after approaching the magnetopause.
The magnetosphere responded to variations in the IMF orientation by magnetic field
perturbations. Magnetic records at ground-magnetometers of INTERMAGNET provided
evidence of a global geomagnetic response in the form of geomagnetic pulses from the equator
to middle latitudes. The earliest response was found at low latitudes in the predawn sector. We
propose a scenario of possible association between dynamical foreshock in the subsolar region,
magnetosheath plasma jets and the deepest injections of the >30 keV electrons at $L < 1.2$ at the
midnight-dawn sector.
*Key words:* trapped energetic electrons, low L-shell, magnetosheath plasma jet, foreshock



**1. Introduction**
Deep injections of tens to hundreds of keV particles into the inner magnetosphere, i.e. drift shells
L < 6, during quiet geomagnetic conditions or weak storm activity have recently become one of
the main issues of radiation belt dynamics (e.g., Turner et al., 2017a; Zhao et al., 2017a). The
cause of "quiet" injections has not been understood yet. An injection depth is estimated using a
notion of drift L-shell, defined by McIlWain (1961). The L parameter determines the unique drift
shell, which remains constant when a charged particle moves adiabatically in the inner
magnetosphere. Numerically, L gives the average geocentric distance to a drift shell at the
magnetic equator. Injection or transport of particles implies violation of adiabatic motion and
changing of L-shell.
The mechanisms responsible for the violation of adiabatic motion of energetic particles are a
subject of extensive modern studies of the radiation belts (e.g., Turner et al., 2015; Turner et al.,
2017b; Zhao and Li, 2013; Zhao et al., 2016; Zhao et al., 2017a). The studies presented some
intriguing challenges for current models of energetic particle injections in L-shell range of 2-6.
Particularly it was pertaining to discrepancy in occurrence frequency, energy range, local time
and penetration depth of electron versus proton injections. Zhao et al. (2016) showed that the
electrons penetrate into the low L-shells more frequently than protons. In addition, it was found
that tens to hundreds of keV electrons penetrate deeper than MeV energy electrons (e.g., Zhao
and Li, 2013; Zhao et al., 2016). It was also found that energetic electrons can often penetrate
down to the slot region separating the inner and outer radiation belts (L ~ 2.5 - 3.5) and also into
the inner radiation belt at L < 2. Moreover, the deepest penetrations of energetic electrons were
revealed even under the inner radiation belt at L < 1.2 (Asikainen and Mursula, 2005; Evans,
1988; Suvorova et al. 2012; 2013).
In the recent study, Zhao et al. (2017a) have compared local time characteristics of electron and
proton flux enhancements in the slot region and suggested that underlying physical mechanisms
responsible for deep penetrations of protons and electrons are different. Particularly, deep proton



penetration is consistent with convection of plasma sheet protons, and deep electron penetration
suggests the existence of a local time localized mechanism. Turner et al. (2015) studied energetic
electron flux enhancements at L < 6 and also suggested that the deep injections at L < 4 (inside
the plasmasphere) may result from a different mechanism than injections observed at higher L
shells (outside the plasmasphere). They hypothesized that the mechanism could be related to
wave activity in the Pi2 frequency range which usually serves as an indicator of substorm
activity. Overall, dynamics of the tens to hundred keV electrons at low L-shells is very different
from dynamics of both protons and electrons at higher L-shells and also in higher energy range.
The ability of energetic electrons to penetrate deeply in the inner zone and below is still puzzling.
An answer to the question may be found by investigating the relation of deep injections of
energetic electrons to solar wind parameters, geomagnetic activity indices and other parameters
of magnetospheric and ionospheric responses (Suvorova, 2017; Zhao et al., 2017b). The studies
mentioned above have reported deep injections of energetic electrons associated with
geomagnetic storms and/or intense substorms, although no significant dependence of penetration
depth or flux intensity on the storm intensity was found (e.g., Suvorova et al., 2013; 2014;
Turner et al., 2017b; Zhao et al., 2016). Some studies noted that deep injections can occur during
nonstorm time but under intense substorm activity (Park et al., 2010; Suvorova et al., 2016;
Turner et al., 2015).
Extensive studies of dynamics of the energetic electrons in the inner radiation belt and below
using the measurements from several satellite missions NOAA/POES, DMSP, DEMETER, and
Van Allen Probes (e.g., Reeves et al., 2016; Suvorova, 2017; Turner et al., 2015, Turner et al.,
2017a; Zhao and Li, 2013; Zhao et al., 2017a) have revealed the following interesting features
such as a high growth rate of fluxes or sudden enhancements, the occurrence of flux
enhancements regardless of storm intensity, the influence of solar wind and geomagnetic
conditions on the occurrence rate, high occurrences of the injections below the inner zone during
specific phases of solar cycles, specific months and local times.



Rapid enhancements of electron fluxes in the inner zone have been known for a long time in
association with deep injections of particles during strong magnetic storms (e.g., Pfitzer and
Winckler, 1968; Imhof et al. 1973; Kikuchi and Evans, 1989; Tanaka et al., 1990). As mentioned,
recent studies showed that rapid or sudden enhancements deep in the inner magnetosphere
cannot be explained by an enhanced convection electric field, convection of plasma sheet
electrons or inward radial diffusion (e.g., Turner et al., 2017b; Zhao et al., 2017a). Increased
statistics have revealed a feature that deep injections may occur frequently, and furthermore,
regardless of storm strength (Tadokoro et al., 2007; Park et al., 2010; Turner et al., 2017a; Zhao
and Li, 2013; Zhao et al., 2016). Another important feature, also mentioned above, is that
injections of the keV electrons and associated flux enhancements can occur even below the inner
belt edge ($L \sim 1.2$), in so-called forbidden zone (Asikainen and Mursula, 2005; Evans, 1988;
Suvorova et al., 2012).
Until recent years, it was believed that these "forbidden injection" events could occur only
during strong magnetic storms and hence could be rarely observed. Note that enhancements in
the forbidden zone were first reported in 1960s (Krasovskii et al., 1961; Savenko et al., 1962;
Heikilla, 1971), however, the conclusions were unconvincing due to the scarce information (see
Paulikas, 1975 for a review). The recent statistical study of electron enhancements in the
forbidden zone showed that the injections below the inner zone can also occur during
geomagnetically quiet conditions (Suvorova, 2017). This fact is consistent with the recent
finding of "quiet" injections in the inner magnetosphere (Turner et al., 2017a; Zhao et al., 2017a).
A case of "quiet" injections of energetic electrons at L < 1.2 is in the focus of our study.
Here, we summarize the main characteristics of the electron injections into the very low L-shells
from several papers (Suvorova and Dmitriev, 2015; Suvorova et al., 2013; 2014; 2016; Suvorova,
2017; Dmitriev et al., 2017). The quasi-trapped energetic electron population in the forbidden
zone, referred to as forbidden energetic electrons (FEE), can be characterized as transient with
highly variable fluxes. The behavior of FEE is similar to keV energy trapped electrons in the



inner radiation belt with flux enhancements in response to magnetic storms (e.g., Kikuchi and
Evans, 1989; Tanaka et al, 1990; Tadokoro et al., 2007; Dmitriev and Yeh, 2008; Zhao and Li,
2013; Selesnick et al., 2016). Simultaneous measurements of particles by satellites at different
altitudes provided clear evidence that the forbidden zone enhancements of energetic electrons
were caused by fast penetration of the inner belt electrons (Suvorova et al., 2014). As known, an
important role in fast transport of particles during storms is played by magnetic and electric field
perturbations. Such perturbations are usually associated with the influence of magnetospheric
substorms, or nighttime processes of magnetic field dipolarizations in the magnetotail (e.g.,
Glocer et al., 2011; Selesnick et al., 2016). However, substorm signatures in the magnetic field in
the low-L region (L< 2) have never been observed.
Thus, the deep injections of keV energy electrons may extend even to the forbidden zone, but
conditions for the fast (~1 - 2 h) earthward transport in the low-*L* region are still unclear.
Nevertheless, the most probable mechanism of the low-*L* injections of energetic electrons was
suggested as the *ExB* drift (e.g., Suvorova et al., 2012), and most of researchers consider and
model an electric drift of electrons in the ExB fields, even though the electric field must be very
high (e.g., Zhao and Li, 2013; Turner et al., 2015; Lejosne and Mozer, 2016; Selesnick et al.,
2016; Su et al., 2016; Zhao et al., 2017a). There is no explanation for penetration of a strong
electric field to such low *L*-shells. What is more important, there is no reliable information on
electric fields at heights of 500-2000 km, because measurements there are difficult, and, as a
consequence of this, empirical electric field models are limited and do not provide the results
below L~2 (e.g., Rowland and Wygant, 1998; Matsui et al., 2013). The most modern research
suggests that the actual strength of penetration electric fields can be stronger than any existing
electric field model at L < 2 (Su et al., 2016).
The studies, mentioned above, have also analyzed a relation between the FEE injections and
geomagnetic activity level. It seemed for a while that intense geomagnetic activity like auroral
substorms was one of the necessary factors for deep electron injections, and the storm-time *Dst-*

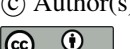



variation did not control the FEE occurrences (Suvorova et al., 2014). It was suggested that
substorm-associated strong electric field can penetrate to the low $L$ region, thereby creating the
conditions for fast earthward transport of trapped electrons in crossed E and B fields. Recent
modeling of the ExB transport mechanism at $L < 1.3$ demonstrated that the mechanism can
successfully operate in the low $L$ region (Selesnick et al., 2016).
However, after that, many FEE events were found during moderate and weak auroral activity,
which was typical for pre-storm (initial phase) or even non-storm conditions (Suvorova and
Dmitriev, 2015; Suvorova et al., 2016). Thus, though no evidence of direct influence of
geomagnetic storms was found, the FEE enhancements appeared to be necessarily associated
with substorm activity in some events studied (Suvorova et al., 2014; 2016). However,
statistically, such a casual relationship with substorms was not confirmed (Suvorova, 2017).
From total statistics of ~530 days with FEE enhancements collected during two solar cycles
(Suvorova, 2017), we found more than three dozen days without essential substorm activity.
These "quiet" events occurred over past decade from 2006 to 2016. The FEE enhancements in
that case were observed only in low energy range of tens of keV.
It is important to mention that one interesting feature was unexpectedly found from the statistical
study (Suvorova, 2017). It is that the most favorable conditions for the FEE enhancements arise
in the period from May to September independently on geomagnetic activity level. A second,
minor peak of occurrence appears in the December - January period. Suvorova (2017) suggested
an important role of the auroral ionosphere in the occurrence of FEE injections. The peculiar
annual variation of the FEE occurrence rate was explained by a change in conductance of the
auroral ionosphere. The conductance depends directly on the illumination of the noon sector of
the auroral zone. As known, the high-latitude ionosphere is better illuminated during solstice
periods, with that the illumination of the northern region is higher than the illumination of the
southern one because of the dipole axis offset relative to the Earth's center. This fact can explain




an existence of two peaks of the FEE occurrence with the major one during the northern summer
period.
The factor of auroral ionosphere conductivity is necessary but not sufficient, and it comes to the
fore during weak geomagnetic activity. External drivers from the solar wind should trigger some
processes in the magnetosphere-ionosphere system that can result in the electron injections into
the forbidden zone. What are these processes when storm and substorm do not develop is still
unclear. A comprehensive analysis of the solar wind drivers and magnetospheric response may
help us to lift the veil. In this paper, we study prominent FEE enhancements during nonstorm
condition on August 1, 2008 in order to determine their possible drivers in the solar wind.

**2. Observations on August 1, 2008**
**2.1. Forbidden Electron Enhancements**
Figure 1 shows large enhancements of the >30 keV electron fluxes at low latitudes on August 1,
2008. The data were compiled from all orbital passes of five NOAA/POES satellites. The
electron fluxes in the energy ranges >30, >100 and >300 keV were measured by the MEPED
instruments boarded on each satellite. The MEPED instrument includes two identical electron
solid-state detector telescopes and measures particle fluxes in two directions: along and
perpendicular to the local vertical direction (Evans and Greer, 2004). The data shown in Figure 1
are from the detector was oriented along the orbital radius-vector, so that it measured quasi-
trapped particles near the equator and precipitating particles in the auroral region. In Figure 1,
the forbidden zone extends in the latitudinal range from -20° to +30° and in the longitudinal
range from 0° to 260°E (or 100°W) that is beyond the South Atlantic anomaly (SAA) at *L<1.2*.
Figure 1a shows the observations of the >30 keV electrons at 0-12 UT, before the enhancements
occurred. Figure 1b shows the interval 12-24 UT, when fluxes of >30 keV quasi-trapped
electrons in the forbidden zone increased by 3 orders of magnitude above a background of ~ $10^2$
$(cm^2 \ s \ sr)^{-1}$ and kept at the enhanced level for several hours. As found previously, the flux



enhancements at low latitudes are peculiar to the quasi-trapped energetic electrons (Suvorova et
al., 2012, 2013). In contrast, enhancements of electrons precipitating at low latitudes are very
rare, weak and short. During the event, precipitating electron fluxes in the forbidden zone did not
increase (not shown). Fluxes of the >100 keV electrons and >30 keV protons did not increase
also (not shown). The quasi-trapped electrons are mirroring at heights below the satellite orbit
(~850 km) in a region of ±30° latitudes, and drift eastward with a rate of 17°-19° per hour toward
the SAA area, where they are lost due to scattering in the dense atmosphere.
Figure 2 and Table 1 present longitudinal and local time locations of 15 FEE enhancements
detected at equatorial passes of POES satellites (P2, P5, P6, P7, P8). Positions of the satellite
orbital planes provided a good coverage of the entire local time (LT) range: 9 - 21 LT (P2 and
P7), 5 - 17 LT (P5 and P6), and 2 - 14 LT (P8). The coverage allows determining the injection
region with uncertainty of approximately 2 h. The first FEE enhancement was observed at ~1250
UT in Central Pacific at night time (2 LT), and the last (enhancement number F15) was detected
at ~2310 UT near the western edge of SAA at day time (17 LT).
It was shown statistically that deep injections into the forbidden zone, similar to plasma sheet
particle injections, occur in the midnight - morning sector (e.g., Suvorova, 2017). During typical
geomagnetic disturbances, nighttime FEE enhancements are observed shortly after local
injections and near an injection site, while subsequent FEE enhancements at daytime are already
the result of azimuthal drift of electrons injected on the nightside. Hence, the nighttime (~2 LT)
enhancements F1 and F4 of >30 keV electron fluxes indicate approximately the time of injection,
respectively, at ~1250 and ~1430 UT or a little bit earlier. After 1530 UT, enhancements were
observed at daytime (numbers F7, F9, and F11-15) and are therefore associated with drifting
electrons.
All remaining enhancements F2, F3, F5, F6, F8 and F10 of >30 keV electron fluxes were
observed in the early morning (5 LT) for a long time interval of ~4 h that lead us to suspect that
the enhancements were observed near the injection site. Nevertheless, we examine the

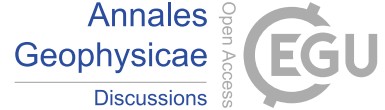



assumption about drift by comparing these enhancements with the injection time for numbers 1
and 4 in Table 1. For the enhancements F1 and F2, 30 keV electrons injected at 1250 UT must
drift ~35.4° of longitude in order to reach the observing satellite P5. It takes ~112 min with the
drift rate of 19°/h for 30 keV electrons at L~1.2 or 125 min with the drift rate of 17°/h at L~1.1.
However, the observed time difference between F1 and F2 is only 25 min that is too short for
drifting from the longitude of F1 to the longitude of F2.
The enhancements F1 and F3 have the longitudinal difference of 26° for 1 h that is much larger
than 19° produced by the drift of >30 keV electrons. Either it could be electrons of slightly
higher energy of ~40-50 keV. However, intensity of these electrons is several times lower than
that for 30 keV electrons because of very steep energy spectrum with maximum in the range of
20-30 keV as shown in the previous study (Suvorova et al., 2013). In contrast, the observations
did not show notable flux decrease. It means that vast majority of the POES/MEPED count rate
is produced by electrons of ~30 keV.
Likewise, one can infer that the enhancement F4 also did not result in the enhancements F5 and
F6 and certainly not in the enhancements F8 and F10. Therefore, the specific longitudinal and
local time distributions of the enhancements indicate multiple injections during about 4.5 h in the
sector of 0 - 6 LT, and the injection region was confined within 3 h of local time over central and
eastern Pacific. In general, these characteristic of injections are in well agreement with those
found from statistics (Suvorova, 2017).

**2.2. Upstream Solar Wind Conditions**
An intriguing aspect of these FEE injection events is that they occurred under quiet, nonstorm
conditions, characterized by Dst/SYM-H ~ 0 nT and AE < 100 nT. We examine solar wind
parameters to search for drivers inducing such deep electron injections. In the study, we focus on
a comparison between the solar wind parameters measured far upstream and near the bow shock





and on their influence on the magnetospheric magnetic field during the period of interest. Global
indices of geomagnetic activity and solar wind data from the Omni high-resolution data set are
shown in Figure 3. The OMNI data base provides solar wind data, which were originally
obtained from upstream monitors (e.g., ACE or Wind satellites) near the L1 libration point at
geocentric distance of ~230 Re (Re is the Earth's radius), and then the data were corrected by
time delay procedure due to propagation to the Earth's bow shock (King and Papitashvili, 2005).
As seen in Figure 3, the solar wind speed and density smoothly varied around averages of 400
km/s and 6 to 4 cm$^{-3}$, respectively, that resulted in gradual change of the dynamic pressure from
2 to 1 nPa. The interplanetary magnetic field (IMF) can be characterized as weakly disturbed by
small-scale structures because of chaotic variations of the magnetic field components and
discontinuities, particularly during the fist half of the day. Also, in this period, the Bz component
was predominately positive. Later, there was a short interval from 1500 to 1800 UT, when IMF
orientation was relatively steady with a continuous negative Bz of about -2 nT. Likely, the
southward IMF resulted in intensification of the *AL* index from 16 to 18 UT with a peak of -250
nT. The 1 min *SYM-H* index was > -10 nT throughout the whole day, indicating there was no
geomagnetic storm. Therefore, the solar wind conditions resulted in a weak auroral disturbance
like an isolated substorm.
Overall, the OMNI magnetic and plasma parameters can be characterized as almost undisturbed
in the period of the FEE enhancements from 1200 to 2300 UT. Obviously, the weak auroral
activity at ~1700 UT could not result in extremely deep injections of the energetic electrons,
which started much earlier, around 1300 UT. Whereas, looking on the PC index, which
represents magnetic activity in the northern (PCN) and southern (PCS) polar caps (Troshichev et
al., 1988), one can see a clear disturbance, particularly in the northern polar cap, in the period
from 1300 to 1530 UT. But it's difficult to identify appropriate solar wind drivers for
interpretation of this polar cap activity.





This raises the question of actual solar wind characteristics at the near-Earth location during the
event. The FEE enhancement event under the nonstorm condition and mild, ordinary solar wind
properties presents intriguing challenge to current understanding of the deep energetic particle
injections, which usually are associated with intense substorm activity. From the characteristic
PC-index behavior, we suspect the actual solar wind parameters affecting the magnetosphere
may be different from those predicted by OMNI. Fortunately, the near-Earth THEMIS mission
can provide necessary reliable information on upstream conditions.

**2.3. THEMIS foreshock observations**
During the time interval from 1200 to 1800 UT, the THEMIS-C satellite (TH-C) had a position
upstream of the bow shock in the subsolar region (Figure 4). The TH-C probe moved from
location (17.2, -0.3, -5.9) Re in GSM at 1200 UT to location (18.1, 3.4, -5.9) Re at 1800 UT.
Hence, we can evaluate characteristics of the upstream solar wind structures actually affecting
the magnetosphere during the period of the FEE enhancements. Figure 5a shows measurements
of the THEMIS-C/FGM fluxgate magnetometer in GMS coordinates with a time resolution of ~3
s (Auster et al., 2008) and the ion spectrograms from THEMIS-C/ESA plasma instrument
(McFadden et al., 2008). The magnetic field components measured in situ by TH-C are
compared with those predicted by OMNI and shown in Figure 5b. Also, Figure 5c presents the
IMF cone angles, between the IMF vector and the Earth-Sun line, for both magnetic data sets.
From 1200 UT to 1320 UT, three TH-C magnetic components demonstrated small-amplitude
variations, and the Bz component had northward direction. During this time, there were
discrepancies between magnetic components of the TH-C and OMNI data caused mostly by time
shift of ~10-15 min, so that TH-C observed arrival of the solar wind structures at earlier time
than that predicted by OMNI. With time correction, one can achieve better consistency in the
two magnetic data sets except the difference in the Bx components about 1310 UT.



In Figure 5c, the OMNI cone angle dropped below 30° between 1330 and 1520 UT that
corresponded to quasi-radial IMF orientation (IMF is almost along the Earth-Sun line), whereas
cone angle variations detected by TH-C were very different from the OMNI data. After 1500 UT,
the OMNI data do not match the TH-C observation any more, even with time correction. About
~1320 UT, ~1400 UT and after 1440 UT, the in-situ observation of THEMIS shows large-
amplitude fluctuations with duration of tens of minutes in three magnetic components and cone
angle (Figure 5a, c). The observed large magnetic fluctuations are ultralow-frequency (ULF)
waves, and they are a typical signature of the upstream region of quasi-parallel bow shocks, so-
called foreshock (e.g., Schwartz and Burgess, 1991). In addition, in the same time intervals, the
plasma spectrogram shows enhancements of suprathermal ion fluxes with energy of >10 keV
(upper panel in Figure 5a). This is another distinguishing signature of the foreshock, known as
diffuse ion population, which is always observed together with the upstream ULF waves
(Gosling et al., 1978; Paschmann et al., 1979; Greenstadt et al., 1980; Crooker et al. 1981).
Hence, the upstream foreshock waves and diffuse ions observed by TH-C in the subsolar region
are associated distinctly with a radial or quasi-radial IMF orientation in the undisturbed solar
wind. Note, that the longest foreshock interval (1435 - 1550 UT) associated with the quasi-radial
IMF orientation was observed by ~20 min later than that predicted by OMNI.
After 1520 UT, the prediction and in-situ data mismatch greatly. The TH-C satellite observed
several rotational discontinuities and alternation between Archimedean spiral and radial
orientations of the IMF vector, while the OMNI magnetic field does not change the Archimedean
spiral orientation from 1520 to 1740 UT. The foreshock returned to the subsolar region
periodically and more frequently in the interval 1600 - 1730 UT than in the earlier period 1320 -
1440 UT.
These two time intervals of frequent foreshock transitions differ in the Bz component: Bz > 0 at
1320 - 1440 UT and Bz < 0 at 1600-1700 UT. It's natural, that the southward Bz results in the
weak auroral activity during the later interval. Nevertheless, the changing direction of IMF has





310 the effect on the magnetic activity in the northern polar cap in the both interval (see the PC index

311 in Figure 1). We check available satellite and ground-based magnetic data to find other responses

312 inside the magnetosphere to the foreshock transitions.


314 **2.4. Magnetospheric magnetic field perturbations**

315 We use magnetic field and plasma measurements in the magnetosphere from the other three

316 THEMIS probes and GOES-12 satellite in order to find signatures of local magnetospheric

317 disturbances. With these data, we examine a magnetospheric response to the subsolar foreshock,

318 which forms each time with arrival of magnetic flux tubes with quasi-radial IMF orientation.

319 Positions of the TH-B, TH-D, TH-E and GOES-12 satellites in the X-Y GSM plane for the

320 period from 1200 to 1800 UT are shown in Figure 4. We used the model of Lin et al. (2010) to

321 calculate magnetopause position. The OMNI data at 1600 UT is used as input data for the model.

322 The GOES12 satellite moved from morning to noon (7 - 13 LT). The TH-E and TH-D probes

323 moved outward from prenoon to postnoon, and the TH-B probe moved inward in the afternoon-

324 dusk sectors.

325 Figure 6 shows variations of the Bz component measured by the TH-E, TH-D, and TH-B probes,

326 the magnetic field strength at geosyncronous orbit (GOES-12), the ion spectrogram from the TH-

327 D satellite and the SYM-H index from 1200 to 1800 UT. As seen in Figure 6 (a, d),

328 characteristics of magnetic field and hot plasma indicate that three THEMIS probes were located

329 inside the dayside magnetosphere during the interval, a region of a strong magnetic field with the

330 magnitude ranging from 40 to 150 nT and low-density of hot (>10 keV) ions. Three THEMIS

331 probes observed significant perturbations in the magnetic field Bz component with

332 increase/decrease of order of several to tens of nT. After 1400 UT, the largest amplitudes were

333 observed by TH-D, which was closer to the magnetopause than other probes at that time (see

334 Figure 4). From 1300 to 1500 UT, there are a few characteristic decreases and enhancements in

335 the Bz component with duration of 20-30 min observed by all probes (Figure 6a). The magnetic



field increases correspond to magnetospheric compressions, and the decreases are
magnetospheric expansions (e.g., Dmitriev and Suvorova, 2012). Prominent magnetic peaks are
indicated by dashed lines and listed in Table 2. At ~1700 and 1715 UT, the TH-D measurements
show that the sign of the Bz component suddenly reversed for a few minutes. The negative Bz
component is a clear signature of the magnetosheath magnetic field. We will consider details of
the magnetosheath intrusion events later.
As seen in Figures 6a-c, THEMIS magnetic observations well correlate with magnetic field
variation observed by GOES-12 and with the SYM-H index in the interval 1300-1600 UT. The
first magnetic pulse was observed at ~13:33:40 simultaneously by TH-B, TH-E, and TH-D and
with a delay of ~2 min by GOES 12. Time moments of magnetic peak 2 coincide for all satellites
(14:20:50 UT). Magnetic peak 3 was observed at first by GOES 12 at ~15:44:00 (~10.6 LT),
then by TH-E at ~15:47:30 (~12 LT) and at last by TH-D at ~15:50:30 UT (~12.5 LT), so a time
difference between GOES 12 and TH-D is ~ 6.5 min and between TH-E and TH-D is 3 min.
The magnetic variations associated with compression-expansion effects could not be caused by
the solar wind pressure variations, which were gradual and small during the interval (see Figure
3). However, the magnetic perturbations may result from local variations in the magnetosheath
pressure. Unfortunately, THEMIS did not measure plasma parameters in the magnetosheath from
1200 to 1600 UT, but an analysis of the later interval (1600-1800 UT) can provide important
information about magnetosheath conditions (see also section 2.5).
After 1545 UT, the TH-D probe observed fast magnetic variations. At that time the probe was
approaching the magnetopause and moving ahead of the TH-E probe (see Figure 4). Note, that
the fast magnetic fluctuations are not always seen in SYM-H and GOES 12 data because of a
low time resolution (1 min) of these data. Figure 6d presents the ion spectrogram from TH-D.
One can see several short-time intrusions of dense and cold plasma with spectrum typical for the
magnetosheath. Moreover, at ~1700 and 1710 UT, the magnetospheric field measured by TH-D
with positive Bz suddenly overturned to negative Bz for a moment that indicated a



362 magnetosheath encounter. Time moments of peaks in the magnetosheath plasma pressure are

363 indicated by lines 4-10 in Figure 6 and listed in Table 2. Below, we analyze characteristics of

364 magnetosheath ions in details.


366 **2.5. Magnetosheath plasma jets interacting with the magnetopause**

367 We analyze the solar wind characteristics in the foreshock region together with the

368 magnetospheric magnetic perturbations and penetration of magnetosheath ions. Figure 7 shows

369 the magnetic field and plasma parameters observed by TH-D, TH-E and TH-C during the

370 interval 1530-1800 UT. In addition, magnetic measurements from GOES 12 and geomagnetic

371 indices are also shown.

372 After 1530 UT, the TH-D and TH-E probes have observed magnetic field pulses associated with

373 the compression effect (Figure 7g). After 1600 UT, TH-D was approaching the magnetopause

374 and started observing occasionally magnetosheath plasma in the magnetosphere, as seen in the

375 ion spectrogram (e.g., lines #4 – 7 and 10, Figures 7b). After 1700 UT, the probe twice entered

376 into and exited from the magnetosheath region as indicated by lines #8 and #9. The

377 magnetosheath plasma can be recognized as dense and cold (<1 keV) ion population. As seen in

378 Figure 7 (panels b and g), not all magnetic pulses are accompanied by plasma penetrations.

379 During the interval, the outermost probe TH-C observed occasionally the foreshock phenomena

380 such as diffuse ions (≥10 keV) in the spectrum (panel a) and large IMF cone angle fluctuations

381 associated with ULF waves (panel h). As one can see, most of the magnetic pulses (panel g)

382 and/or magnetosheath ion populations (panel b) indicated by lines #3, #4, and #6-10 (i.e. except

383 #5) were accompanied by the foreshock diffuse ions (panel a).

384 Figure 8 shows characteristics of magnetosheath plasma in details for three intervals 1600-1630,

385 1630-1700, and 1658-1728 UT. Since plasma charge neutrality means equal density of ions and

386 electrons, Figure 8 presents parameters of the ion component only (panels a-d). Dynamic





pressure and density of the solar wind plasma measured far upstream by the ACE monitor are
also shown for comparison in panels (b, c). The time period from 1600 to 1630 UT is shown in
panels (a1-g1). The probes TH-D and TH-E observed magnetic field variation in specific
depletion-hump sequence from 1607 to 1614 UT (panels f1, g1), similar to the variations
indicated by lines #1 - #3 in the earlier interval (see Figure 6). Magnetic peak is indicated by line
#4. Additionally, wave-like structures with a period of ~30-60 sec (in the ULF range) are clearly
seen in magnetic measurement of both probes during the time interval from 1609 to 1627 UT
(panels f1, g1). At 1614 - 1616 UT, TH-D observed cold ions (~100 eV - 3 keV) and electrons
(<1 keV, not shown) of the magnetosheath origin staying in the magnetosphere (panel a1). The
plasma has maximal speed of >200 km/s and high density of 3-9 cm$^{-3}$ that result in the high total
pressure of 1.5 - 1.8 nPa (panels b1-d1). Its dynamical characteristics distinctly exceed the solar
wind parameters with density of 4 - 5 cm$^{-3}$ and total pressure of ~1.1 nPa (panels b1, c1). Internal
structure of plasma forms 3 prominent pressure pulses between 16:14:50 and 16:16:00 UT, a
central pulse is dominated by magnetic component (panel f1) and two lateral pulses are
dominated by dense plasma components (panel c1). Two plasma density enhancements produced
a diamagnetic effect seen as a characteristic decrease of magnetic field (panel f1). At the outer
edge of the plasma structure, the anti-sunward velocity (Vx < 0) reached high value of -100 km/s,
indicating that the local plasma flow struck and interacted with the magnetopause (panel d1).
The Vz component demonstrates a maximal value in southward direction (-200 km/s). Three
rotated velocity components Vx, Vy and Vz indicate that vortex-like plasma structure propagated
along the magnetopause toward south and dusk. This dense and high-speed plasma structure is
analogous to the large-scale magnetosheath plasma jet studied by Dmitriev and Suvorova (2012).
Large-scale magnetosheath plasma jets are defined as intense localized fast ion fluxes whose
kinetic energy density is several times higher than that in the upstream solar wind and duration is
longer than 30 sec (Dmitriev and Suvorova, 2015).



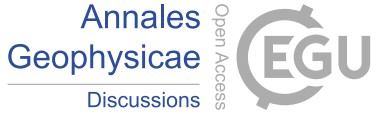

Panels (a2-g2) in Figure 8 show magnetic compressions and magnetosheath penetrations (lines
#5 - #7) during the time period from 1630 to 1700 UT. It is also seen that the magnetic field
measured by TH-E was disturbed by ULF wave activity (panel g2). The plasma structures #5 and
#6 (panel a2) have short durations and are characterized by extremely high density of 16 and 12
cm$^{-3}$, respectively, that well explain the compression effects in magnetic measurements from TH-
E and TH-D (panels f2, g2). Prolonged plasma structure #7 has lower density of 4 - 9 cm$^{-3}$ and
did not produce a notable compression effect in accordance with to TH-E magnetic
measurements (panel g2). It is important that inside each plasma structure, we reveal a dense
plasma core, which is characterized by enhanced speed of ~150 or ~220 km/s with a dominant
Vz component (negative or positive). These parameters, typical for plasma jets, formed pressure
of high magnitude, which exceeded the upstream solar wind pressure by 50-80 % (panel b2).
Likely, magnetosheath plasma jets interacted with the magnetopause, and then they were
partially trapped thereby penetrating into the magnetosphere (Dmitriev and Suvorova, 2015).
The amount of this penetrated plasma estimated by the authors can be comparable with estimates
of the total amount of plasma entering the dayside magnetosphere (Sibeck, 1999).
During the last time period 1658 - 1728 UT shown in panels (a3-g3), we have an excellent
opportunity to examine plasma parameters in the magnetosheath region adjacent to the
magnetopause. Panels (a3-f3) show two cases of magnetopause distortions followed by short
intervals of the magnetosheath from ~1700 to 1701 UT and from 1711 to ~1715 UT. The ULF
wave activity is also clearly seen in the magnetic measurement of the TH-E probe (panel g3).
The TH-D probe at distance of ~10.8 Re and ~13 LT suddenly crossed the magnetopause and
moved into the magnetosheath, a region where the magnetic field vector rotated to negative Bz
(panel f3). Plasma in both magnetosheath intervals has extremely high density (~20 cm$^{-3}$) and
high velocity (≤ 200 km/s). In the magnetosheath, one can see local pressure pulses around
~1700 UT and ~1712 UT (lines #8 and 9). For #9 case, TH-E observed a small shallow hump of
the magnetic field of a few nT between two depletions at 1707 and 1715 UT (panel g3). The last



event (#10) shown in Figure 8c is a short penetration of magnetosheath plasma accompanied by
a small perturbation in the magnetospheric field observed at ~1724-1725 UT (panels e3, f3).
Density and pressure of this structure did not exceed the solar wind parameters, though the
velocity was large (~150 km/s) with dominant negative Vz component (panel b3-d3).
Thus, we found typical characteristics of dense and fast plasma jets in all intrusions of the
magnetosheath plasma into the magnetosphere and in the magnetosheath itself. Most of these
structures caused local compression effects at the dayside. Also, the TH-E magnetic field is
modulated by ULF waves in the range of magnetic pulsations Pc 3-4 with period between 10 and
60 seconds. As known, dayside Pc 3-4 waves are originated in the upstream solar wind and
penetrate into the magnetosphere, while their amplitude is controlled by a foreshock position or
IMF orientation (e.g., Guglielmi, 1974).
As shown in Figure 3, moderate auroral and polar cap activity was observed during the same
time (1600-1800 UT). However, it should be noted that in the preceded interval 1300-1600 UT,
associated with the deep electron injections and FEE enhancements, the THEMIS probes also
observed similar magnetic compression-expansion effects at inner part of orbits (~7 - 10 Re). At
that time, we found enhanced magnetic activity in the polar cap (only in the northern
hemisphere), but no auroral activity. This raises an interesting question about spatial pattern of
geomagnetic field response to the impact of magnetosheath pressure pulses/plasma jets
interacting probably with the dayside magnetopause in the earlier interval 1300-1600 UT with
magnetic enhancements #1- #3.

**2.6. Global ground-based magnetic variations**
The global dynamics of geomagnetic field perturbations were studied using 1-min magnetic data
provided by an INTERMAGNET of ground magnetometers (http://www.intermagnet.org/index-
eng.php). Since there were no pressure pulses in the upstream solar wind and auroral activity was





low (see Figure 3), we expect that variations in the geomagnetic field (if any) should result from
the local magnetosheath pressure pulses. We used magnetic stations located at geomagnetic
latitudes below ~60° (Table 3), where a significant effect of different propagation time of
magnetohydrodynamic (MHD) waves in the magnetosphere will be almost hidden at 1 min
resolution. We grouped magnetic stations in meridional and latitudinal chains.
Figure 9 presents relative variations of horizontal (H) component, which was measured at
equatorial and low latitudes ranging from 0° to ~20° of geomagnetic latitude in the interval from
1100 to 1600 UT. In Figure 9, the stations are arranged in local time from morning to
postmidnight. The THEMIS magnetic field measurements are also shown at bottom. Four
magnetic field pulses of different amplitudes are seen around ~1200, ~1335-1345, ~1422-1430
and ~1545-1550 UT practically at all stations. The last three pulses correspond to those observed
by THEMIS at ~1334, ~1421 and 1547-1550 UT (#1 - #3, see also Table 2). Moreover, one can
see the same pattern of magnetic variation "enhancement and decrease" in both ground-based
and satellite observations. Note that the first magnetic pulse at ~1200 UT can not be emerged
from THEMIS data because of the large background magnetic field in the inner magnetosphere.
Magnetic records at daytime and nighttime are clearly distinguished by amplitudes and time
delay relatively to the THEMIS data.
Magnetic records at nighttime stations (PHU, GZH, KNY, KDU, GUA, HON, PPT) are
characterized by prominent variations of H component, with peak-to peak amplitudes of 3 - 5 nT.
The dayside stations (KOU, VSS, MBO, ASC, TSU, BNG, AAE, ABG) show relative weak, but
still distinguished, magnetic humps. Smaller amplitude at daytime is a result of an amplifying
integral effect from the Chapman-Ferraro current at the magnetopause and ionospheric Sq-
current at the ground.
It is interesting, that the magnetic pulse at 1200 UT is simultaneously (within the accuracy of ~1
min resolution) observed in all local time sectors. However, the other three enhancements were
observed in different LT sectors at slightly different time. A time difference varies from ~2 min



to ~10 min. The time delay depends on the time moment when a jet interacts with the
magnetopause in a given latitude-longitude sector (Dmitriev and Suvorova, 2012).
We draw attention to the fact that low-latitude HON and PPT stations, which were located in the
predawn sector (2-5 LT) from 1300 to 1500 UT, demonstrate the best coincidence (with a delay
of ~1 min) of magnetic enhancements #1 and #2 with those observed by THEMIS near noon.
Nighttime and daytime stations (PHU, GZH, KNY, KDU, GUA, MBO, ASC, TSU, BNG, AAE,
ABG) observed these peaks with ~3 - 5 min delay. The longest delay (~7 min) for pulses #1 and
#2 is found at morning/prenoon stations KOU and VSS (~9 - 11 LT).
As we have showed above, the FEE injections (F1 - F6 in Table 1) occur from ~2 to 5 LT. So,
we present meridional chains of stations in the predawn and midnight sectors (Figure 10). All
magnetic enhancements are well recognized from 0° to 60° of geomagnetic latitude. In midnight
and predawn sectors, the first magnetic pulse at ~1200 UT was observed practically
simultaneously everywhere. Magnetic pulse #1 around ~1333 UT was delayed by ~7 min at
midlatitudes (30°-60°) in the midnight sector (left panel) and by ~5 min in the predawn sector
(right panel). The pulse #2 shows a smaller delay (~3 min) at midlatitudes. The magnetic pulse
#3 at most stations in both sectors is observed around ~1545 UT, that is 2 min earlier than at TH-
E and 1 min later than at GOES (see Table 2). Thus, the low and middle latitude geomagnetic
observations in all local time sectors demonstrate that the magnetic variations of "enhancement-
decrease" pattern at 1200-1600 UT were observed by ground magnetometers as a global
phenomenon.

**3. Discussion and Summary**
In this work, using NOAA/POES and THEMIS satellites we investigated an unusual case of
deep injections of >30 keV electrons at L< 1.2 and associated FEE enhancements occurred
during quiet, nonstorm condition on August 1, 2008. A series of night injections of >30 keV



electrons could be associated with transient magnetospheric magnetic field perturbations. These
magnetic perturbations were observed globally like "compression-expansion" effects by
THEMIS and GOES 12 in the magnetosphere and by most of ground-based magnetometers from
INTERMAGNET network. Comparative analysis of the THEMIS, OMNI and ACE data showed
that the magnetic perturbations were caused by impact on the magnetopause by a series of
plasma pressure pulses propagated through the magnetosheath but not in the undisturbed
upstream solar wind. Such plasma jets are typical consequence of the foreshock dynamics driven
by variations in the IMF orientation (e.g., Lin et al., 1996) and are comprehensively studied
using THEMIS and MMS missions (e.g., Archer et al., 2012; 2013; Dmitriev and Suvorova,
2012; 2015; Plaschke et al., 2017). For our case, THEMIS measurements in the region in front of
the bow shock, showed obvious evidences of transient quasi-parallel bow shock and foreshock
conditions during the interval.
The strong FEE enhancements of intensity $\sim 10^{4\text{-}5}$ $(cm^2\ s\ sr)^{-1}$ were observed by POES in central
and eastern Pacific for a long time from ~1300 to 2300 UT. With analysis of longitudinal and
local time distributions of the enhancements we identified a series of night injections occurring
occasionally in the sector of 2-5 LT in the period from ~1300 to ~1700 UT (Figure 2). We found
that the injections of >30 keV electrons F1 - F6 (Table 1) occurred at much earlier time than the
weak auroral activity (1600 - 1800 UT), and hence, were unlikely related to it. The injections F8
and F10 occurred during the weak auroral activity interval. Also in that time, THEMIS-D
approached the magnetopause and detected magnetosheath plasma intrusions into the dayside
magnetosphere.
The quiet geomagnetic conditions in the period of 1300 - 1600 UT are consistent with
undisturbed solar wind conditions obtained from the OMNI data and ACE upstream monitor that
is not surprising. However, the picture, emerged from the THEMIS-C magnetic observations
right upstream of the subsolar bow shock at ~19 Re, showed an apparent discrepancy with
OMNI in the magnetic field structures (see Figure 5). For our case, the discrepancy appeared to





be due to an inability to predict accurately the evolution of small-scaled structures with quasi-
radial magnetic tubes during the propagation to the Earth and, as result, a notable uncertainty in
the time lag method applied in the OMNI database. Erroneous time lag is typical for cases of the
quasi-radial IMF orientation (e.g., Case and Wild, 2012; Mailyan et al., 2008; Bier et al., 2014;
Suvorova and Dmitriev, 2016). The actual solar wind parameters affecting the magnetosphere
were found to be related to a subsolar foreshock, which was observed by THEMIS. The analysis
of the THEMIS observations helps us to recognize possible external drivers, which might be
responsible for the deep FEE injections.
It is important to emphasize that only with OMNI data or with any far upstream monitor (ACE,
Wind, etc.), it would be impossible to resolve this unusual event. First, the OMNI data, as
mentioned above, present the upstream data modified by the timing procedure. But a problem of
the accuracy of time delays still exist as noted by a number of authors (e.g., Case and Wild, 2012;
McPherron et al., 2013; Mailyan et al., 2008). For example, the propagation time of magnetic
structures is determined less accurately for radial IMF condition (e.g., Bier et al., 2014;
Borovsky, 2008; Suvorova and Dmitriev, 2016). Second, there is a high probability that small-
scale magnetic field structures observed far upstream evolve unpredictably during the
propagation toward the Earth, so that the resulting structure can be different (e.g, Zastenker et al.,
2000; Borovsky, 2008). For example, the interval 1600 - 1800 UT is a good illustration for this,
proving no similarity exists in IMF cone angle variations in the solar wind and foreshock regions
(see Figure 5). Fortunately, we have a possibility to use more reliable information from the near
earth THEMIS mission on solar wind during the time period of interest.
During the period 1200 - 1800 UT, the magnetosphere was periodically under the quasi-radial
IMF conditions (Figure 5). The response of the magnetic field to these conditions was studied
with THEMIS located in the solar wind and magnetosphere and with ground-base
magnetometers of INTERMAGNET network. During the quasi-radial IMF intervals, THEMIS
observed intense ULF activity in the foreshock region. It is well known that the foreshock is also





accompanied by ULF waves observed inside the magnetosphere by satellites and ground based
magnetometers (e.g., Guglielmi, 1974; Clausen et al., 2009; Bier et al., 2014). However, the
amplitude of those magnetospheric ULF waves seems not strong enough to result in anomalous
radial transport of energetic electrons at $L < 1.2$.
The THEMIS measurements in the magnetosphere clearly show several local effects of
compression and expansion in the interval 1200 - 1600 UT (#1 - #3 in Table 2), and
magnetosheath plasma penetrations and magnetosheath encounters in the interval 1600 - 1800
UT (#4 - #10 in Table 2). The earlier interval, which is associated with several FEE injection
events, was investigated using ground magnetometer records. Signatures of magnetic variations
similar to the THEMIS observations were found in the H component at majority of ground
stations located from the geomagnetic equator to midlatitude (Figures 9 and 10). Common
feature for three magnetic pulses is that they were observed first at low latitudes in the
postmidnight/predawn sector (2-5 LT), and then their local time and latitudinal patterns become
quite different and complicated. Thus, the geomagnetic field responded globally to the local
pressure impacts compressing the dayside magnetosphere. At that, the postmidnight/predawn
sector (2-5 LT) shows the earliest pronounced response at low latitudes.
Analysis of the later interval 1600 - 1800 UT (Figure 7) indicated a possible cause of the
magnetic variations. Note that upstream bow shock conditions observed by TH-C during both
time intervals were similar in that the quasi-radial IMF appeared. During that time, THEMIS (D,
E) observed magnetic pulses, some of which were accompanied by penetrations of
magnetosheath plasma into the magnetosphere. THEMIS also encountered the magnetosheath
for a few minutes. Dense and high-speed plasma jets or pressure pulses were found in all plasma
structures, which penetrated into the magnetosphere or propagated in the magnetosheath (Figure
8). Obviously, impact of these magnetosheath plasma jets on the dayside magnetopause caused
compression effects in the magnetospheric field. This interval was accompanied by two FEE





injections F8 and F10 at 1633 and 1712 UT, respectively, which followed the magnetic pulses 4
- 8 occurred from 1614 to 1700 UT.
The magnetosheath pressure pulses or plasma jets arose during time intervals when quasi-radial
IMF tubes were passing the subsolar bow shock region as observed by THEMIS. The foreshock
was occasionally moving in or out of the subsolar region (see Figure 5). As the spacecraft
crossed an interface between two flux tubes, it observed a rotation discontinuity. Passages of the
rotational discontinuities followed by change between quasi-parallel and quasi-perpendicular
bow shock regimes created favorable conditions for generation of plasma jets (Lin et al., 1996).
Note that jets can be generated by directional discontinuities in absence of foreshock conditions
(cases #1 and #5) (Dmitriev and Suvorova 2012). THEMIS was able to observe directly such
plasma jets in the magnetosheath at later time, when it approached closely the magnetopause.
Similar effects of transient magnetospheric compression and expansion and their signatures at
low-latitudinal ground magnetometers were studied by Dmitriev and Suvorova (2012, 2015). As
they established, such magnetic field perturbations were caused by a pressure pulse impact of
magnetosheath plasma jet striking the dayside magnetopause during a foreshock transition
through the subsolar region toward flank.
Another important effect is penetration of the magnetosheath plasma into the magnetosphere at
low latitudes due to interaction of large-scale jets with the magnetopause (Dmitriev and
Suvorova, 2015). Recently, it was revealed that the magnetosheath high-speed jets result in
auroral brightening on the dayside (Han et al., 2017a; Wang et al., 2018). Sometimes, the aurora
penetrates to lower latitudes, so-called throat aurora. Han et al. (2017a) found that quasi-radial
IMF or subsolar foreshock condition is favorable for occurrence of dayside throat aurora,
whereas southward IMF has a weaker influence on its occurrence. Based on the comprehensive
study of properties of throat aurora, Han et al. (2018) concluded that throat auroras are definite
ground signatures for local magnetopause deformations and compressions produced by
magnetosheath plasma jet impact. They also provided direct evidence that the source of



precipitating particles in the throat auroras was the magnetosheath plasma (sometimes mixed
with magnetospheric plasma), which was effectively transported by jets from the magnetosheath
(Han et al., 2016; Han et al., 2017b). Thus, the jet impact is responsible for generating throat
aurora, which provides enhancements in auroral ionospheric conductivity on the dayside.
Dmitriev and Suvorova (2015) have found that the average rate of jet-related penetration of the
magnetosheath plasma into the magnetosphere is about $10^{29}$ particles per day. The penetrated hot
ions with energies of ~1 keV move quickly (within a few minutes) along the magnetic field lines
to high-latitude regions of the dayside ionosphere. We can estimate the flux of precipitating ions
of ~$10^7$ to $10^8$ (cm$^2$ s)$^{-1}$ if we assume that particles precipitate on the dayside arc of 3º width at
70º latitude. Hence, we can assume that those ions can produce significant additional ionization
and increase conductivity of the high-latitude ionosphere on the dayside that induces an
enhancement of the electric field on the nightside and especially in the predawn sector, where the
conductivity is weak. The nightside electric field might penetrate from high to low latitudes and
produce ExB drift of electrons from the inner radiation belt to lower heights.
Thus, we can propose a scenario when magnetosheath plasma jets, associated with dynamical
subsolar foreshock and rotational discontinuities, interact with the dayside magnetopause and
cause compression effect with magnetic field perturbations and effective transport of the
magnetosheath plasma inside the magnetosphere. The magnetosheath plasma or mix with
magnetospheric plasma precipitates to the dayside ionosphere at high latitudes that result in a
local increase of the ionospheric conductivity. This in turn promotes generation of transient
localized electric fields, which are able to penetrate from high latitudes to very low latitudes (low
L-shells). Most favorable conditions for penetration of localized electric fields and FEE
enhancements arise in the period from May to September independently on geomagnetic activity
level (Suvorova, 2017). Our case event on 1 August 2008 corresponds well to these favorable
conditions.





Anomalous transport and loss of energetic particles in the magnetosphere was studied and
modeled in numerous papers (e.g., Glocer et al., 2011; Selesnick et al., 2016; Su et al., 2016;
Turner et al., 2015; Turner et al., 2017a; Zhao and Li, 2013; Zhao et al., 2017a). In the present
case, the magnetosphere is driven rather by plasma jets generated locally in the magnetosheath.
Moreover, we show that the solar wind conditions right upstream of the bow shock can be
substantially different from those measured in the far upstream regions. Another serious problem
is the generation/penetration of electric fields in the inner magnetosphere, which is far from
complete understanding. Numerical estimations show that the anomalous (fast) radial transport
of particles observed in the inner magnetosphere can be produced by the electric field up to 5
mV/m (Selesnick et al., 2016; Suvorova et al., 2013). At the present time, there are no models
predicting strong electric fields in the inner radiation belt and below. In this sense the scenarios
suggested here requires further development of new advanced models of the magnetosheath –
magnetosphere – ionosphere coupling.

**Acknowledgements**

The authors thanks a team of NOAA's Polar Orbiting Environmental Satellites for providing experimental data about energetic particles, the CDAWEB for providing the ACE solar wind data and Kyoto World Data Center for Geomagnetism (http://wdc.kugi.kyoto-u.ac.jp/index.html) for providing the geomagnetic indices. We thank the THEMIS team for magnetic and plasma data provided. The results presented in this paper rely on data collected at magnetic observatories. We thank the national institutes that support them and INTERMAGNET for promoting high standards of magnetic observatory practice (www.intermagnet.org). The OMNI data are provided by the GSFC/SPDF OMNIWeb platform (http://cdaweb.gsfc.nasa.gov/). The work was supported by grant MOST 106-2811-M-008-050 and MOST 106-2111-M-008-030-MY3 to National Central University.

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





**Table 1** *FEE Enhancements observed by POES satellites*

| FEE ID # | POES s/c ID | Observed time hh:mm UT | Longitude deg | LT* h |
|---|---|---|---|---|
| F1 | P8 | 12:50 | -164.2 | 1.8 |
| F2 | P5 | 13:15 | -128.8 | 5.1 |
| F3 | P6 | 13:53 | -138.3 | 5.1 |
| F4 | P8 | 14:32 | 169.7 | 1.6 |
| F5 | P5 | 14:54 | -152.7 | 5.1 |
| F6 | P6 | 15:34 | -162.5 | 5.0 |
| F7 | P2 | 15:44 | -98.7 | 9.3 |
| F8 | P5 | 16:33 | -170.1 | 5.0 |
| F9 | P7 | 16:37 | -107.3 | 9.7 |
| F10 | P6 | 17:12 | 180.0 | 4.9 |
| F11 | P2 | 17:24 | -123.0 | 9.4 |
| F12 | P7 | 18:16 | -131.0 | 9.8 |
| F13 | P2 | 19:06 | -140.0 | 9.6 |
| F14 | P8 | 20:30 | -105.0 | 13.8 |
| F15 | P6 | 23:09 | -94.5 | 17.2 |

* Local time



**Table 2** *Timing of Magnetic Field Enhancements and Plasma Pulses from THEMIS and GOES12*

| ID # | s/c ID | Time of magnetic peak hh:mm:ss | Time of pressure pulse hh:mm:ss | Foreshock signatures |
|---|---|---|---|---|
| 1 | TH-D | 13:33:40 | | absent |
| | TH-E | 13:33:40 | | |
| | TH-B | 13:33:40 | | |
| | G12 | 13:35:40 | | |
| 2 | TH-D | 14:20:50 | | ions, ULF |
| | TH-E | 14:20:50 | | |
| | TH-B | 14:20:50 | | |
| | G12 | 14:20:50 | | |
| 3 | TH-D | 15:50:30 | | ions, ULF |
| | TH-E | 15:47:30 | | |
| | G12 | 15:44:00 | | |
| 4 | TH-D | 16:14:05 | ~16:15 - 16:16 | ions, ULF |
| | TH-E | 16:14:05 | | |
| 5 | TH-D | 16:38:20 | 16:40 | absent |
| | TH-E | 16:38:40 | | |
| 6 | TH-D | 16:47:45 | 16:47:55 | ions |
| | TH-E | 16:47:45 | | |
| 7 | TH-D | - | 16:51:22 | ions, ULF |
| | TH-E | - | | |
| 8 | TH-D | magnetosheath | 17:00:25 | ions, ULF |
| | TH-E | - | | |
| 9 | TH-D | magnetosheath | ~17:12 - 17:13 | ions |
| | TH-E | 17:12:30 | | |
| 10 | TH-D | 17:24:50 | 17:24:50 | ions, ULF |
| | TH-E | - | | |





**Table 3**

*Location of Magnetic Stations in Geographic and Geomagnetic coordinates*

| Code | Name | GLat[a] | GLon[a] | MLat[b] | MLon[b] |
|------|------|------|------|------|------|
| AAE | Addis Ababa | 9.0 | 38.8 | 5.3 | 109.9 |
| ABG | Alibag | 18.6 | 72.9 | 9.5 | 144.4 |
| ASC | Ascension Island | -8.0 | -14.4 | -1.4 | 54.7 |
| ASP | Alice Springs | -23.8 | 133.9 | -34.1 | -153.6 |
| BNG | Bangui | 4.3 | 18.6 | 4.6 | 89.3 |
| CMO | College | 64.9 | -147.9 | 64.8 | -102.6 |
| CNB | Canberra | -35.3 | 149.4 | -43.8 | -134.5 |
| CTA | Charters Towers | -20.1 | 146.3 | -29.1 | -140.7 |
| EYR | Eyrewell | -43.4 | 172.4 | -47.8 | -107.0 |
| GUA | Guam | 13.6 | 144.9 | 4.2 | -146.3 |
| GZH | Zhaoqing | 23.0 | 112.5 | 11.7 | -177.1 |
| HON | Honolulu | 21.3 | -158.0 | 21.2 | -92.7 |
| KAK | Kakioka | 36.2 | 140.2 | 26.2 | -153.3 |
| KDU | Kakadu | -12.7 | 132.5 | -23.2 | -156.3 |
| KNY | Kanoya | 31.4 | 130.9 | 20.7 | -161.2 |
| KOU | Kourou | 5.2 | -52.7 | 16.1 | 17.7 |
| MBO | Mbour | 14.4 | -17.0 | 21.1 | 55.8 |
| MCQ | McQuarie Island | -54.5 | 159.0 | -60.9 | -116.2 |
| MMB | Memambetsu | 43.9 | 144.2 | 34.2 | -150.9 |
| PET | Paratunka | 53.0 | 158.3 | 45.6 | -138.5 |
| PHU | Phuthuy | 21.0 | 106.0 | 9.7 | 176.0 |
| PPT | Pamatai | -17.6 | -149.6 | -15.2 | -76.5 |
| SHU | Shumagin | 55.4 | 199.5 | 54.1 | -103.1 |
| SIT | Sitka | 57.1 | -135.3 | 60.1 | -83.7 |
| TSU | Tsumeb | -19.2 | 17.6 | -18.3 | 83.5 |
| VSS | Vassouras | -22.4 | -43.7 | -12.1 | 24.6 |

[a] Geographic latitude and longitude

[b] Magnetic latitude and longitude



FIGURE CAPTIONS

**Figure 1.** Geographic distribution of >30 keV electron fluxes measured by five NOAA/POES satellites on August 1, 2008 for the time interval (a) 0-12 UT, before the electron flux enhancements and (b) 12-24 UT, during the enhancements. In the forbidden zone, at low latitudes and equator, the quasi-trapped electron fluxes enhanced largely during nonstorm condition after 12 UT. The forbidden zone is bounded by L=1.2 (white lines) and located outside of the South Atlantic Anomaly (SAA) at equatorial-to-low latitudes. The solid black curve indicates the dip equator.

**Figure 2.** Locations of FEE enhancements in longitude and local time (black circles). Measurements within the SAA area are indicated by the open circles. Colorful curves denote low-latitude orbital passes of five NOAA/POES satellites: P2 (black), P5 (pink), P6 (red), P7 (blue), and P8 (green).

**Figure 3.** Solar wind parameters from OMNI data and geomagnetic indices on August 1, 2008. From top to bottom: (a) solar wind density (black) and dynamic pressure (blue), (b) solar wind speed, (c) interplanetary magnetic field (IMF) components Bx (blue), By (green), Bz (red) and magnitude B (black) in Geocentric Solar Magnetospheric (GMS) coordinates, (d) polar cap magnetic activity index PCN for northern (blue) and PCS for southern (red) hemispheres, (e) auroral electrojet index AE (black), AL (red), AU (green), and (f) storm time ring current variation index SYM-H. The shaded box denotes the time interval from 13 to 23 UT, when the nonstorm FEE enhancements were observed.

**Figure 4.** Spacecraft positions in GSM coordinates from 1200 to 1800 UT on August 1, 2018. The TH-C probe (blue) was in front of the subsolar bow shock. The TH-E (orange), TH-D (green), TH-B (brown), and GOES 12 (black) were located inside the dayside magnetosphere. The magnetopause position (black curve) was calculated using OMNI data for the upstream conditions at ~1600 UT following the model by Lin et al.'s (2010).

**Figure 5.** Observations of plasma and magnetic field on August 1, 2008. (a) Ion spectrogram (ion flux is in units of eV/cm$^2$ s sr eV) and IMF vector components in GSM coordinates measured by TH-C, (b) IMF vector components from OMNI data set, (c) IMF cone angles plotted for TH-C (red) and OMNI (black).

**Figure 6.** Satellite measurements of magnetic field and plasma in the dayside magnetosphere and geomagnetic activity. (a) The Bz-GSM components from THEMIS probes TH-B (brown), TH-E (orange), and TH-D (green). The left y-axis corresponds to the magnetic measurements from TH-B and TH-D, and the right y-axis to TH-E. (b) The magnetic field strength from GOES-12 (black); (c) the SYM-H index; and (d) the ion spectrogram from TH-D (ion flux is in

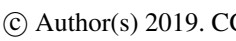


units of eV/cm$^2$ s sr eV). Dashed lines, numbered from 1 to 10, indicate time moments of magnetic and plasma disturbances observed by THEMIS.

**Figure 7.** Observations of plasma and magnetic field at 1530-1800 UT on August 1, 2008: (a-c) ion spectrograms measured by TH-C, TH-D, and TH-E (ion flux is in units of eV/cm$^2$ s sr eV), (d) SYM-H index, (e) AE (black) and AL (red) indices, (f) horizontal magnetic field Hp detected by GOES 12 from 10 to 13 LT, (g) magnetic field strengths Btot from TH-D (green) and TH-E (red), (h) IMF cone angles for TH-C (black) and for the ACE upstream monitor (blue). The ACE measurements are delayed by 60 min. Dashed lines and numbers #4 - #10 mark magnetospheric disturbances with magnetosheath ion population observed in the magnetosphere.

**Figure 8.** Observations of plasma and magnetic field during the intervals 1600 - 1630 UT, 1630 - 1700 UT and 1658 - 1728 UT on August 1, 2008. Panels show from top to bottom: (a) ion spectrogram from TH-D, (b) total pressure Ptot measured by the ACE upstream monitor (black) and TH-D (red), (c) plasma density D measured by ACE (black) and TH-D (blue), (d) TH-D measurements of bulk velocity V (black) and its components in GSM coordinates Vx (blue), Vy (green) and Vz (red), (e) transversal components of magnetic field Bx (blue) and By (green) from TH-D, (f) magnitude B and Bz component of magnetic field from TH-D, (g) magnitude B and Bz component of magnetic field from TH-E. The magnetosheath plasma penetration is denoted by dashed lines and numbers #4 - #10.

**Figure 9.** Relative variations in the horizontal component (H) of the geomagnetic field at low geomagnetic latitudes. Local time intervals are indicated near the station codes. The vertical lines depict time of the magnetic pulses at THEMIS (lines #1 - #3). Bottom panel shows magnetic field B measured by TH-E (orange) and by TH-D (green).

**Figure 10.** Relative variations in the horizontal component (H) of the geomagnetic field in the midnight (left) and predawn (right) sectors. The geomagnetic latitudes of the stations are indicated near station codes. The vertical lines depict time of the magnetic pulses at THEMIS.

Ann. Geophys. Discuss., https//doi.org/10.5194/angeo-2019-5


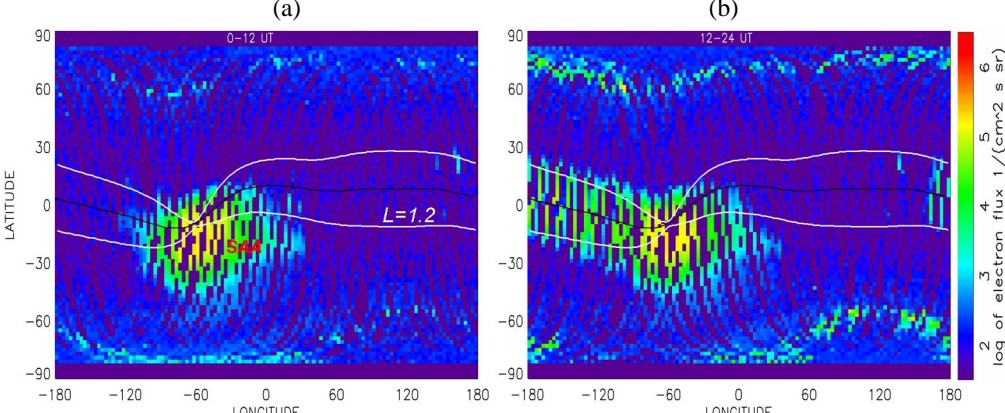

**Figure 1.** Geographic distribution of >30 keV electron fluxes measured by five NOAA/POES satellites on August 1, 2008 for the time interval (a) 0-12 UT, before the electron flux enhancements and (b) 12-24 UT, during the enhancements. In the forbidden zone, at low latitudes and equator, the quasi-trapped electron fluxes enhanced largely during nonstorm condition after 12 UT. The forbidden zone is bounded by L=1.2 (white lines) and located outside of the South Atlantic Anomaly (SAA) at equatorial-to-low latitudes. The solid black curve indicates the dip equator.





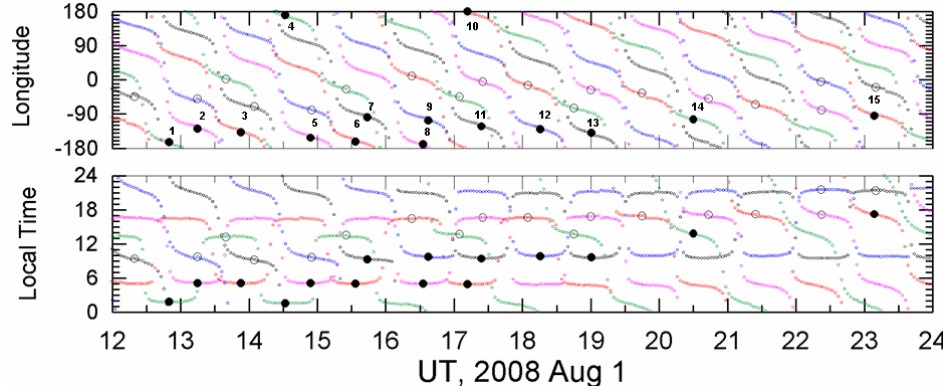

**Figure 2.** Locations of FEE enhancements in longitude and local time (black circles). Measurements within the SAA area are indicated by the open circles. Colorful curves denote low-latitude orbital passes of five NOAA/POES satellites: P2 (black), P5 (pink), P6 (red), P7 (blue), and P8 (green).



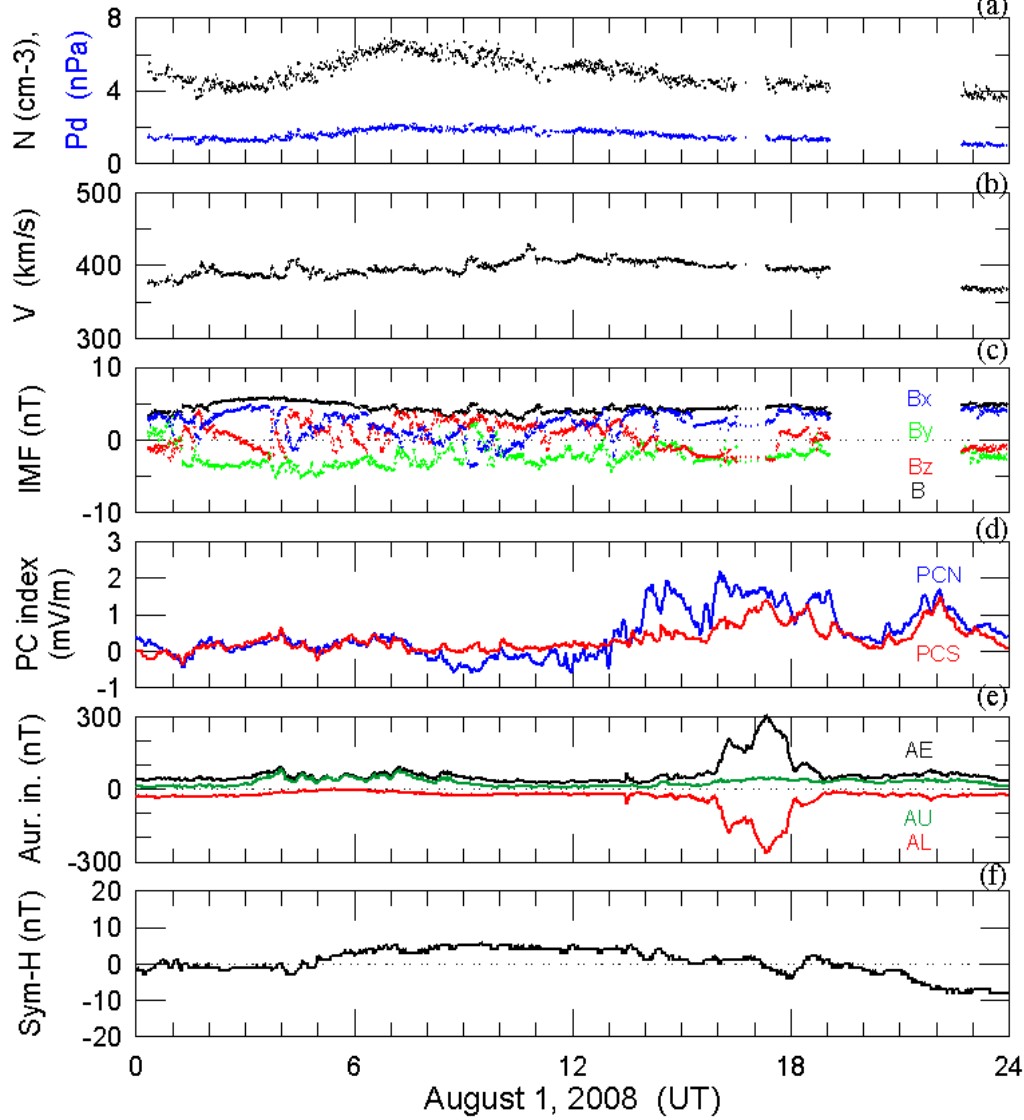

**Figure 3.** Solar wind parameters from OMNI data and geomagnetic indices on August 1, 2008. From top to bottom: (a) solar wind density (black) and dynamic pressure (blue), (b) solar wind speed, (c) interplanetary magnetic field (IMF) components Bx (blue), By (green), Bz (red) and magnitude B (black) in Geocentric Solar Magnetospheric (GMS) coordinates, (d) polar cap magnetic activity index PCN for northern (blue) and PCS for southern (red) hemispheres, (e) auroral electrojet index AE (black), AL (red), AU (green), and (f) storm time ring current variation index SYM-H. The shaded box denotes the time interval from 13 to 23 UT, when the nonstorm FEE enhancements were observed.



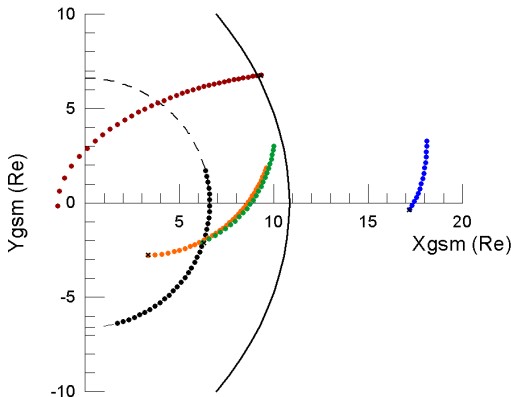

**Figure 4.** Spacecraft positions in GSM coordinates from 1200 to 1800 UT on August 1, 2018. The TH-C probe (blue) was in front of the subsolar bow shock. The TH-E (orange), TH-D (green), TH-B (brown), and GOES 12 (black) were located inside the dayside magnetosphere. The magnetopause position (black curve) was calculated using OMNI data for the upstream conditions at ~1600 UT following the model by Lin et al.'s (2010).



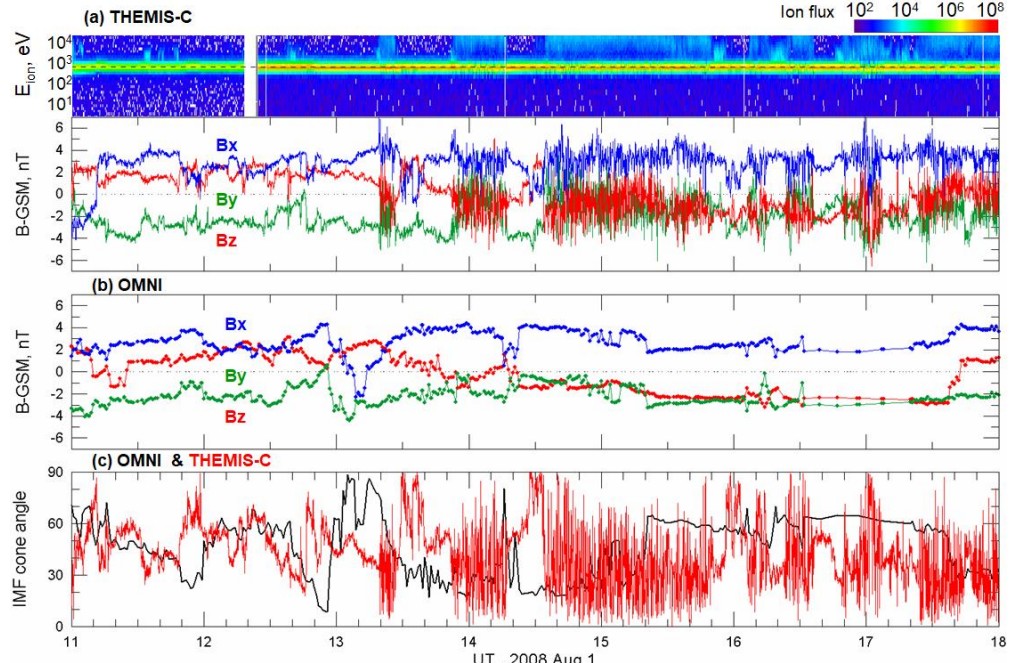

**Figure 5.** Observations of plasma and magnetic field on August 1, 2008. (a) Ion spectrogram (ion flux is in units of eV/cm$^2$ s sr eV) and IMF vector components in GSM coordinates measured by TH-C, (b) IMF vector components from OMNI data set, (c) IMF cone angles plotted for TH-C (red) and OMNI (black).



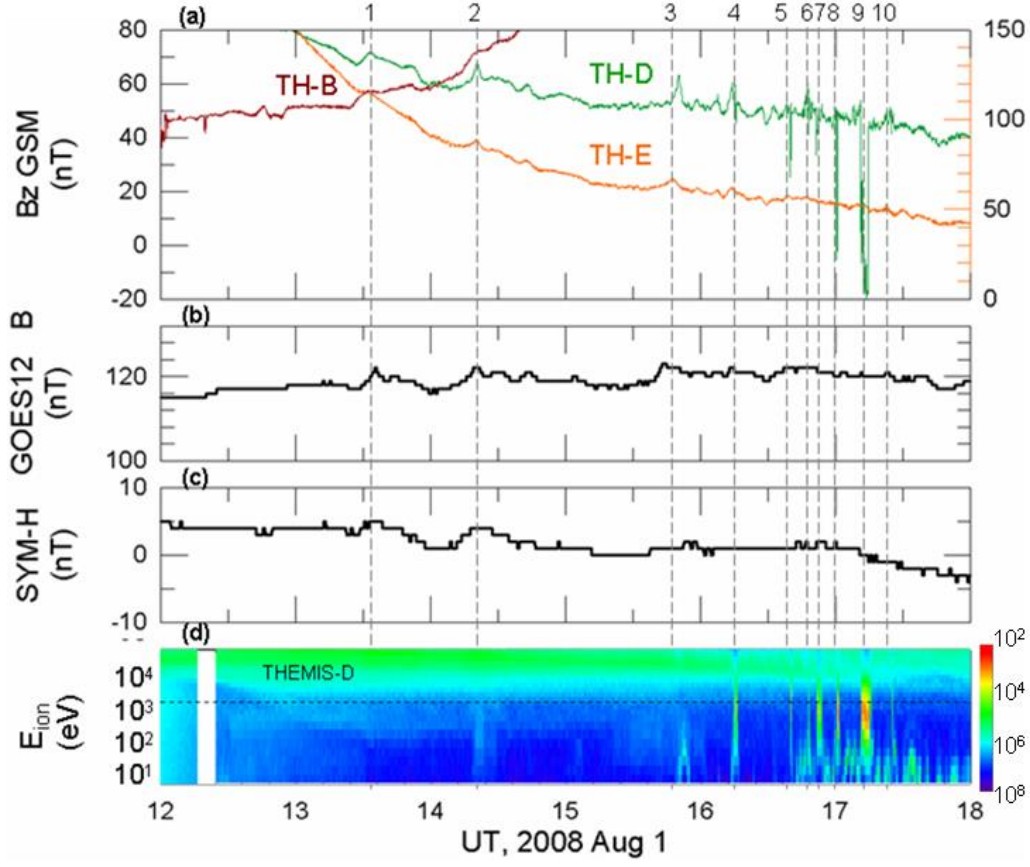

**Figure 6.** Satellite measurements of magnetic field and plasma in the dayside magnetosphere and geomagnetic activity. (a) The Bz-GSM components from THEMIS probes TH-B (brown), TH-E (orange), and TH-D (green). The left y-axis corresponds to the magnetic measurements from TH-B and TH-D, and the right y-axis to TH-E. (b) The magnetic field strength from GOES-12 (black); (c) the SYM-H index; and (d) the ion spectrogram from TH-D (ion flux is in units of eV/cm$^2$ s sr eV). Dashed lines, numbered from 1 to 10, indicate time moments of magnetic and plasma disturbances observed by THEMIS.




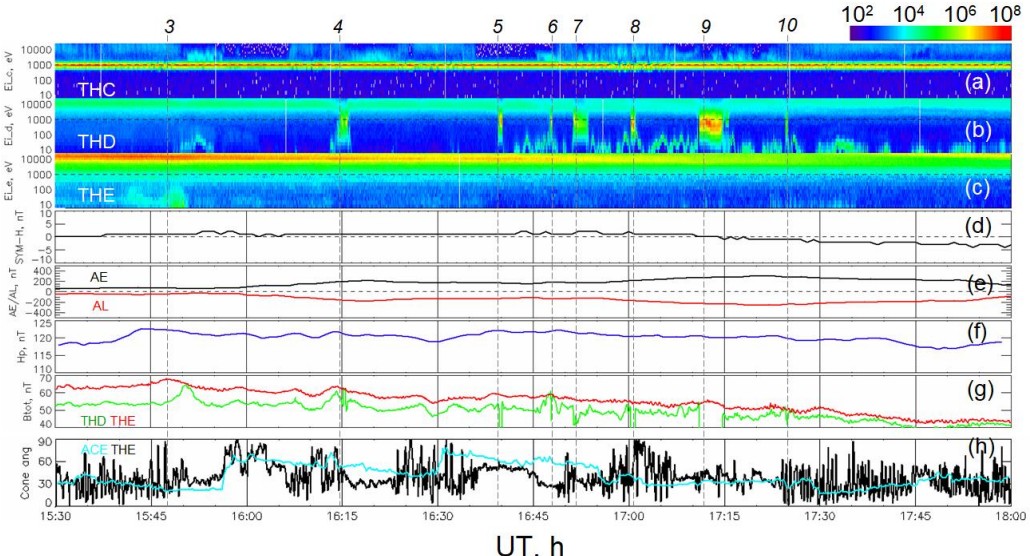

**Figure 7.** Observations of plasma and magnetic field at 1530-1800 UT on August 1, 2008: (a-c) ion spectrograms measured by TH-C, TH-D, and TH-E (ion flux is in units of eV/cm$^2$ s sr eV), (d) SYM-H index, (e) AE (black) and AL (red) indices, (f) horizontal magnetic field Hp detected by GOES 12 from 10 to 13 LT, (g) magnetic field strengths Btot from TH-D (green) and TH-E (red), (h) IMF cone angles for TH-C (black) and for the ACE upstream monitor (blue). The ACE measurements are delayed by 60 min. Dashed lines and numbers #4 - #10 mark magnetospheric disturbances with magnetosheath ion population observed in the magnetosphere.





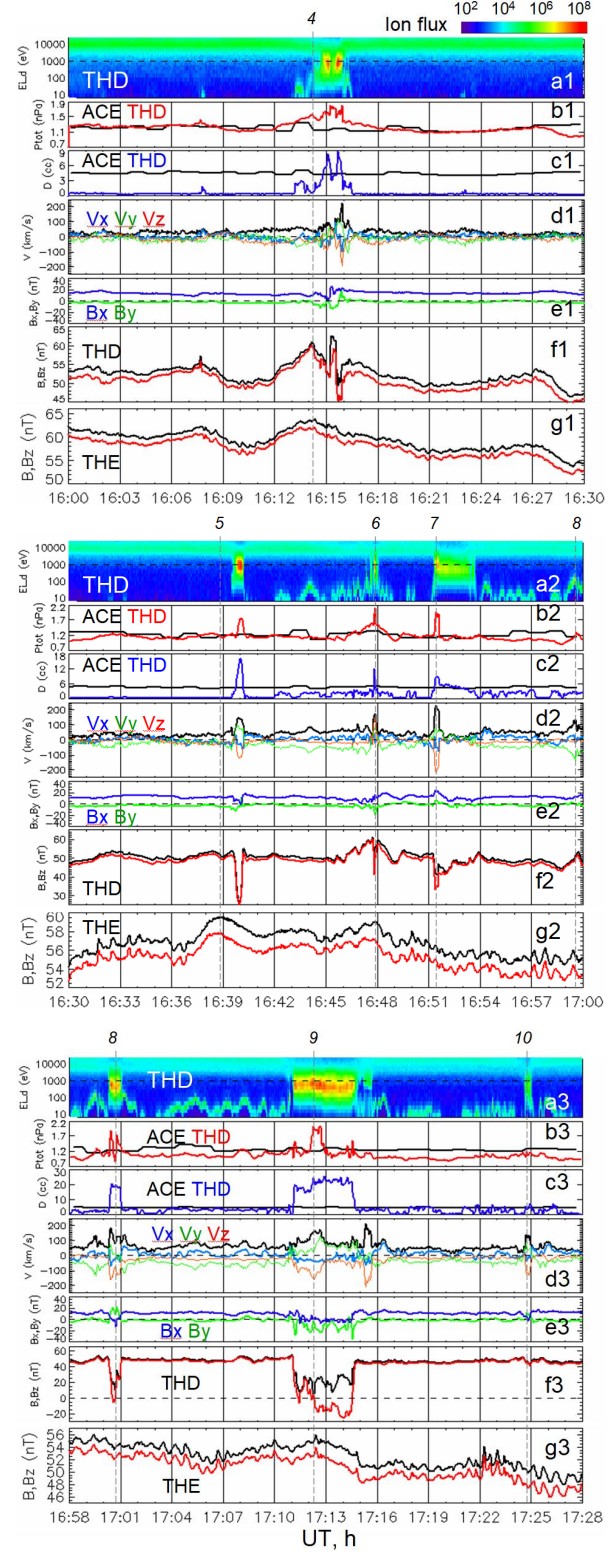





**Figure 8.** Observations of plasma and magnetic field during the intervals 1600 - 1630 UT, 1630 - 1700 UT and 1658 - 1728 UT on August 1, 2008. Panels show from top to bottom: (a) ion spectrogram from TH-D, (b) total pressure measured by the ACE upstream monitor (black) and TH-D (red), (c) plasma density measured by ACE (black) and TH-D (blue), (d) TH-D measurements of bulk velocity V (black) and its components in GSM coordinates Vx (blue), Vy (green) and Vz (red), (e) transversal components of magnetic field Bx (blue) and By (green) from TH-D, (f) magnitude B and Bz component of magnetic field from TH-D, (g) magnitude B and Bz component of magnetic field from TH-E. The magnetosheath plasma penetration is denoted by dashed lines and numbers #4 - #10.



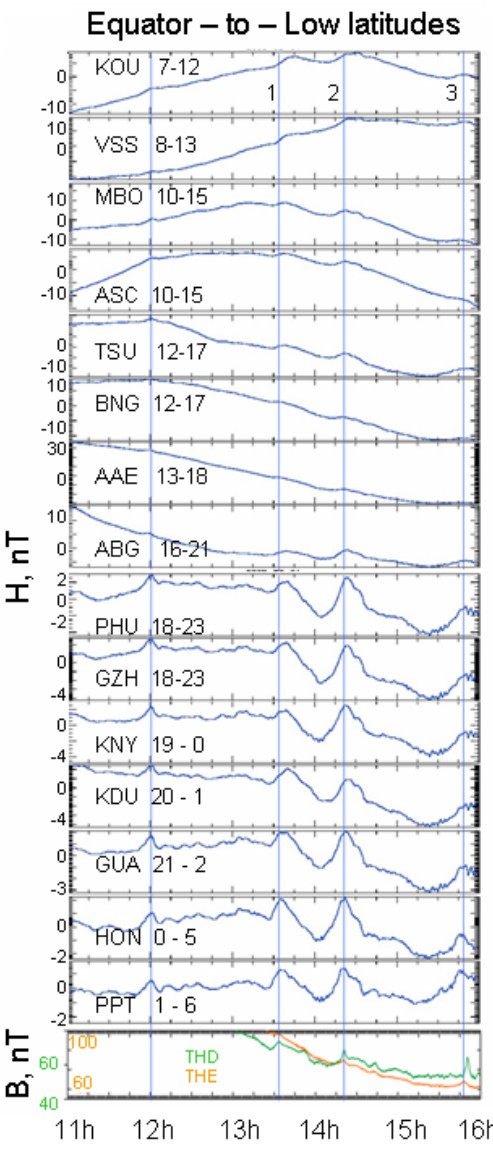

**Figure 9.** Relative variations in the horizontal component (H) of the geomagnetic field at low geomagnetic latitudes. Local time intervals are indicated near the station codes. The vertical lines depict time of the magnetic pulses at THEMIS (lines #1 - #3). Bottom panel shows magnetic field B measured by TH-E (orange) and by TH-D (green).




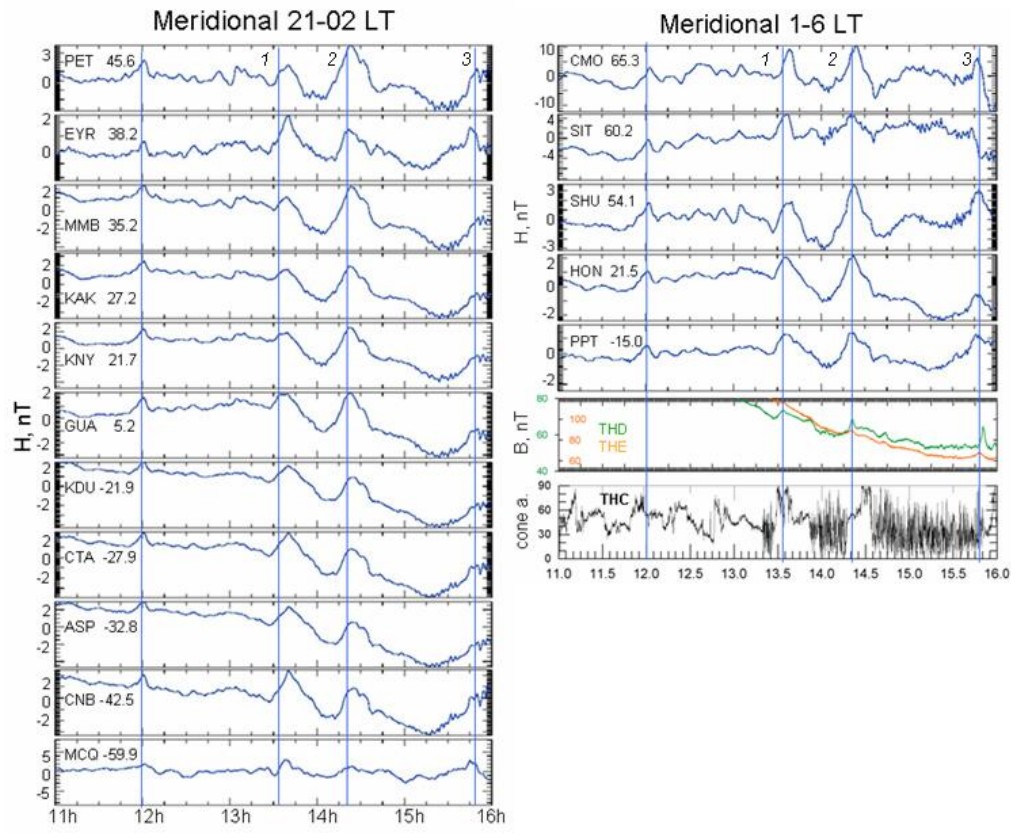

**Figure 10.** Relative variations in the horizontal component (H) of the geomagnetic field in the midnight (left) and predawn (right) sectors. The geomagnetic latitudes of the stations are indicated near station codes. The vertical lines depict time of the magnetic pulses at THEMIS.