# Peer review of "Energetic electron enhancements under radiation belt (L < 1.2) during"

_Annales Geophysicae, 2019_

## Referee Comment (RC1) · Anonymous Referee #1 · 19 Feb 2019

General Comments:

This manuscript reports a series of >30 keV electron flux enhancement events that happened at L<1.2 observed by POES satellites, and massive related observations from THEMIS, ground magnetometer, ACE, etc. These events are likely to be a subset of the events analyzed in Suvorova (2017) and this study is a follow-up work related to Suvorova (2017). In the present study, the authors propose that the magnetic perturbation near the magnetopause causes a mixture of magnetosheath plasma and magnetospheric plasma to precipitate in high latitude (high L regions) which further induce a large transient electric field that could transport the electrons to L<1.2. However, there

is no solid evidence reported to prove that the flux enhancements at L<1.2 are caused by magnetic perturbation near the magnetopause, nor analysis on the possibilities that this proposed chain of processes could work. The reviewer suggests to at least add in some more solid arguments or simulation results to prove that the proposed processes are reasonable before the paper can be published. The reviewer also suggests the authors to be more concise on some part of the paper, to avoid extra confusions of the readers.

Specific Comments:

1. The authors presented the >30 keV electron flux measurements by POES satellites in Figure 1. In Figure 1, it is clear that electron fluxes are enhanced in the quasi-trapped region (outside of SAA), but the fluxes in SAA that are more stably-trapped almost remain the same. The authors refer to those events as injections in many places in the paper (e.g., line 202, 208). However, if those electrons are injected from higher L, they are supposed to become more 90 degree peaked in pitch angle, which means they are more likely to be stably-trapped and more enhancements in the SAA region are expected. From Figure 1, the slot region is not filled, which is supposed to be seen in an typical injection event that penetrates down to L=1.2. In fact, previous studies such as Li et al (2017, titled "Measurement of electrons from albedo neutron decay and neutron density in near-Earth space") reported events that enhanced stably-trapped electrons are observed due to geomagnetic activities while the quasi-trapped electron fluxes stay the same. Moreover, people would easily link the enhancements in the quasi-trapped electrons to enhanced pitch angle scattering. The authors should show more detailed observations of these events and explain why these events are injections. The author should also specify the looking direction of the detector in the caption of Figure 1.

2. As is stated in the general comments, the authors have not present any solid evidence that the electron enhancements at L=1.2 could be caused by magnetic pertur-bations near the magnetopause which is at quite large L. Only coincidences in time are

shown in the present study. The reviewer suggests to show more solid arguments or some simulation results to prove this possibility. In Li et al (2017), which is mentioned above, they also state that the large electric field can only cause an L shell distortion of 0.01 and this process is energy-dependent. Please comment on it and the possibility that the electric field moves the electrons to L<1.2 in this case.

Here are some other comments:

3. In Table 1, the authors list a series of flux enhancement events observed by POES. The author should specify the criteria used to select those events, and show some detailed electron flux profile of those events, such as how long the enhancements last, specific L shell of each event or how many data points are included in each event. The reviewer also suggests to use more commonly used names for POES satellites such as POES-15/18... instead of P2/P5...

4. Line 210-227: the authors intend to prove that each flux enhancement event is individual and not caused by any other event, for example, F2 is not caused by F1. However, this analysis is based on the presumption that the event is really transient. The authors should show some evidence to argue such as F1 could not have been enhanced 100 min before the observation of F1. Also, please explain why this is important. The reviewer does not find it very essential to the analysis later and suggests to be more concise on this problem.

5. Line 227: Please specify if these events are a subset of Suvorova (2017) event list. If so, the authors should make a clarification before stating that the characteristics agree with those in Suvorova (2017), otherwise it is misleading.

6. Figure 3: Since the authors show that L1 is not a preferable location for observations of the magnetic perturbations as compared to Themis, this figure is not necessary. The reviewer suggests to combine some of the information in Figure 3 into Figure 4 and be more concise on the text as well, in order to help the readers to focus on the important part, Themis observations.

7. Line 611: Please use explicit number of the latitude of throat aurora instead of "lower latitude" here. It is misleading because this study is talking about phenomena at L=1.2 (<30 deg in latitude), while the throat aurora in a series of Han et al papers is still located at >70 deg in latitude (or please correct this number).

---

## Short Comment (SC1) · 21 Feb 2019

Dear Referee1, Thank you very much for your comments and suggestions. Before to address all your concerns, we feel necessary to explain some crucial points regarding the manuscript.

"In the present study, the authors propose that the magnetic perturbation near the magnetopause causes a mixture of magnetosheath plasma and magnetospheric plasma to precipitate in high latitude (high L regions) which further induce a large transient electric field that could transport the electrons to L<1.2. However, there is no solid evidence reported to prove that the flux enhancements at L<1.2 are caused by magnetic per-

turbation near the magnetopause, nor analysis on the possibilities that this proposed chain of processes could work." "2. As is stated in the general comments, the authors have not present any solid evidence that the electron enhancements at L=1.2 could be caused by magnetic perturbations near the magnetopause which is at quite large L."

We want to clarify this point because it is important for overall understanding of our study. We did not state that the observed magnetic perturbations near the magnetopause or inside the outer magnetosphere, at large L, can cause such a mixture of plasma at high latitudes and electron enhancements at the equator. These perturbations had small amplitudes of about of several to tens of nT [e.g., line 332]. We wrote [line 513] "A series of night injections of >30 keV electrons could be associated with transient magnetospheric magnetic field perturbations." We wrote about the association in other parts of text also. These magnetic perturbations are only a response of the geomagnetic field on occasional pressure pulses produced by magnetosheath plasma jets at the magnetopause. In the study, we emphasized an important role of transient subsolar foreshock condition, under which plasma jets are generated, for the magnetosphere–ionosphere coupling, particularly for non-storm events. The transient subsolar foreshock was only recently recognized as a major driver for a throat aurora, as we mentioned in the study.

The second important moment, which we want to clarify in this comment, concerns the paper Li et al. "Measurement of electrons from albedo neutron decay and neutron density in near-Earth space" mentioned by Referee. In particular, the Referee puts attention that Li et al. "state that the large electric field can only cause an L shell distortion of 0.01 and this process is energy-dependent. Please comment on it and the possibility that the electric field moves the electrons to L<1.2 in this case".

First of all, that study is about relativistic electrons (∼500 keV) during a geomagnetic storm. We think it has a little relation to our study of low-energy (>30 keV) electrons during quiet nonstorm conditions. Moreover, we note that Li et al. have cited the studies by Selesnik et al (2016) and Su et al. (2016). In the text we also discussed

results of these studies. These studies compared observations of electron injections below L=2 with simulations. Su et al. demonstrated that "an enhanced large-scale electric field can be responsible for injection of ∼100 keV electrons in the inner radiation belt." They also noted that "it is thus not necessary for electrons to be transported all the way from the outer zone during a single injection." Selesnik et al. investigated a more deep injections at L<1.2 for energies 100-400 keV. They wrote "Injection to L<1.2 is demonstrated in both observations and simulations by the end of 23 June, but the simulated injection is smaller because the model Ec (electric field) was reduced to zero for L<1.17". The both studies pointed out that none of the existing electric field models can accurately describe the penetration field and, hence, deep injections (L<1.2). Su et al. (2016) : "An accurate global electric field model is a necessary requirement in order to correctly capture the nondiffusive radial transport in the inner radiation belt." Our paper presents new experimental results, which help to develop a new model. The new model should be a subject of another paper.

Sincerely, Alla Suvorova

---

## Short Comment (SC2) · 22 Feb 2019

Specific Comment 1 by Referee 1

(1.1) The authors presented the >30 keV electron flux measurements by POES satellites in Figure 1. In Figure 1, it is clear that electron fluxes are enhanced in the quasitrapped region (outside of SAA), but the fluxes in SAA that are more stably-trapped almost remain the same.

Reply:

Actually, electron fluxes did increase in the SAA region. However, the background

fluxes in SAA were already high (several units of 10ˆ5 (cm2 s sr)-1). The fluxes of FEE were mostly less than 10ˆ5 (cm2 s sr)-1. Hence, they produce a little increase of the flux in SAA which is hard to be seen in the logarithmic scale. However, this effect is beyond the scope of our study.

(1.2) The authors refer to those events as injections in many places in the paper (e.g., line 202, 208). However, if those electrons are injected from higher L, they are supposed to become more 90 degree peaked in pitch angle, which means they are more likely to be stably-trapped and more enhancements in the SAA region are expected. From Figure 1, the slot region is not filled, which is supposed to be seen in an typical injection event that penetrates down to L=1.2. In fact, previous studies such as Li et al (2017, titled "Measurement of electrons from albedo neutron decay and neutron density in near-Earth space") reported events that enhanced stably trapped electrons are observed due to geomagnetic activities while the quasi-trapped electron fluxes stay the same. Moreover, people would easily link the enhancements in the quasi-trapped electrons to enhanced pitch angle scattering. The authors should show more detailed observations of these events and explain why these events are injections.

Reply:

In order to clarify this crucial issue we revised Figure 2 (see below) in order to show the time profiles of intensities and L-shells of FEE enhancements. We revised the text accordingly: "Figure 2 and Table 1 present main characteristics of 15 FEE enhancements detected along equatorial passes of POES satellites (P2, P5, P6, P7, P8). We analyze the peak fluxes in the FEE enhancements (time, local time, longitude, and L-shell). " "As seen in Figure 2a,b, the FEE enhancements peak at minimal L-shells, i.e. at the equator. The fluxes decrease quickly with growing L. This pattern corresponds to a fast radial transport (injection) of electrons from the inner radiation belt. Note that pitch-angular scattering of electrons gives different profiles: the fluxes should be minimal and the equator and grow with L-shell."

Concerning to the albedo neutron mechanism. It is impossible to apply this mechanism to the FEE enhancements because of the following well known facts: (1) The fluxes of albedo neutrons at equatorial latitudes are much lower (order of magnitudes) than the fluxes of FEE. (2) During magnetic quiet, the latitudinal profile of secondary particles (generated in decay) is positive, i.e. the flux of secondary particles increases with latitude due to a decrease of the cut-off rigidity of incident cosmic rays. Figure 2 demonstrates totally different pattern.

(1.3) The author should also specify the looking direction of the detector in the caption of Figure 1.

Reply:

In the caption of Figure 1 we add the following sentences: "The electrons are detected in vertical direction. In the forbidden zone, those electrons are quasi-trapped."

Caption of revised Figure 2:

FEE enhancements on 1 August 2008: (a) fluxes of >30 keV electrons in units (cm2 s sr)-1, (b) L-shell of enhancements, (c) longitude and (d) local time of peak fluxes (black circles). Measurements within the SAA area are indicated by the open circles. Colorful curves denote NOAA/POES satellites: P2 (black), P5 (pink), P6 (red), P7 (blue), and P8 (green). Horizontal dashed line at panel (b) depicts the lower edge of the inner radiation belt. FEE enhancements peak at the equator (minimal L-shells) that indicates a fast radial transport from the inner radiation belt.

Flux

(a)

L-shell

(b)

Longitude

(c)

Local Time

(d)

UT, 2008 Aug 1

**Fig. 1.** revised Figure 2

---

## Short Comment (SC3) · 22 Feb 2019

Specific Comment 3 by Referee 1

(3) In Table 1, the authors list a series of flux enhancement events observed by POES. The author should specify the criteria used to select those events, and show some detailed electron flux profile of those events, such as how long the enhancements last, specific L shell of each event or how many data points are included in each event. The reviewer also suggests to use more commonly used names for POES satellites such as POES-15/18...instead of P2/P5...

[Figure]

Reply:

In the revised Figure 2, the intensity and L-shells of enhancements are shown. In the revised manuscript, we specify the criteria more precisely:

". . . the forbidden zone extends at L < 1.2 in the latitudinal range from -20° to +30° and in the longitudinal range from 0° to 260°E (or 100°W) that is beyond the South Atlantic anomaly (SAA). . . . Figure 1b shows the interval 12 - 24 UT, when fluxes of >30 keV quasi-trapped electrons in the forbidden zone increased by 3 orders of magnitude above a background of ~10ˆ2 (cm2 s sr)-1 and kept at the enhanced level for several hours. We have selected FEE enhancements with intensity >10ˆ3 (cm2 s sr)-1. "

We think that abbreviation P2, P5 etc. are more convenient for presentations in Figures and Tables.

Specific Comment 4 by Referee 1

(4) Line 210-227: the authors intend to prove that each flux enhancement event is individual and not caused by any other event, for example, F2 is not caused by F1. However, this analysis is based on the presumption that the event is really transient. The authors should show some evidence to argue such as F1 could not have been enhanced 100 min before the observation of F1. Also, please explain why this is important. The reviewer does not find it very essential to the analysis later and suggests to be more concise on this problem.

Reply:

In the original manuscript we have already clarified this important issue: "Figure 1a shows the observations of the >30 keV electrons at 0 - 12 UT, before the enhancements occurred."

In the revised manuscript we provide an additional explanation:

"At that time, the satellites passed the same regions but they did not detect any FEE

enhancements."

The suggestion of multiple injections is important because several injections are accompanied by several jets. We find correspondence between the jets and injections. Note that there were no substorm-associated injections in the present case.

---

## Referee Comment (RC2) · Anonymous Referee #2 · 25 Feb 2019

General Comments

This paper reports energetic (>30 keV) electron flux enhancement at L<1.2 measured by the NOAA/POES satellites and relate it to the transient injection of magnetosheath plasma into the dayside magnetopause region, which is measured by the THEMIS satellite, and global geomagnetic pulses, which are measured by ground INTERMAG-NET magnetometers and GOES satellites. The authors propose a scenario of possible association between these dayside magnetopause phenomena with the deep injection of >30keV electrons at L<1.2 by the penetration of localized electric field.

The electron flux enhancement at L<1.2 is well described including its research history

which is very interesting. Looking through this paper, however, I think the connection between the observed phenomena occurring in the dayside magnetosheath / magnetopause region and the electron flux enhancement at L<1.2 is weak and not well validated by the observations reported in this paper. These two phenomena occur in the same half day of 12-24 UT on August 1, 2008. But there is a significant possibility that they occur in the same day "by chance". I think it is necessary to provide some more concrete evidence including some quantitative estimation that can explain the observed L<1.2 electron enhancement.

Specific Comments

1. The descriptions of OMTI, THEMIS, GOES and ground magnetometers are fair and easy to understand, although they can be shorter. The authors propose a scenario that dayside magnetopause phenomena cause magnetosphere compression, and associated magnetosheath / magnetospheric plasma precipitation to the dayside ionosphere at high latitudes that result in a local increase of the ionospheric conductivity. This in turn promotes generation of transient localized electric fields, which are able to penetrate from high latitudes to very low latitudes to accelerate energetic electrons at L<1.2. However, in the nightside auroral zone we have normal aurora and associated ionospheric conductivity change which can be much larger than those in the dayside aurora. If the scenario proposed by the authors works, why we do not have L<1.2 acceleration during ordinary (non-storm time) substorms which occur almost every day and cause strong aurora and associated conductivity change in the nightside high latitudes? If L<1.2 electron flux enhancement does not occur during ordinary substorms, I think it indicates that the proposed scenario does not work in the actual magnetosphere.

2. As shown in Figures 7 and 8 the THEMIS satellites shows repeating motion in and out from the magnetosphere to the magnetosheath. Such in and out features are very often seen when THEMIS is approaching to the magnetopause region, because the magnetopause location is not fixed and changes due to dynamic pressure change in
the magnetosheath and/or surface waves caused by Kelvin-Helmholz instability in the magnetopause. In the present case, since compressional wave signatures are seen in GOES and ground magnetometers, it is likely that the dynamic pressure variation outside the magnetosphere is the cause of this motion of THEMIS in/out from the magnetosphere. But I think such compressional wave with an amplitude of a few nano-tesla is not unusual and occur frequently. Then how often does the authors find L<1.2 electron acceleration? Is this a frequent phenomenon occurring associated with the frequently-occurring compression of the magnetosphere with the amplitude of a few nano-tesla on the ground magnetometers? How the authors can prove that these two phenomena occurs in the same time not by chance? Maybe the authors can check correspondence of timing between each magnetospheric compression and the electron flux enhancement at L<1.2.

3. The authors show magnetic field pulses observed by GOES and ground magne-tometers. If the penetrating electric field is propagating in the magnetosphere, it should be related to the observed magnetic field variations by the Maxwell's equation of dB/dt = -rot E. One can argue that the observed magnetic field variation (dB/dt) can be used to estimate electric field by taking only one component of the rotation, e.g., dB/dt = dEx/dy (dEx = dB/dt * dy). The GOES magnetic field amplitude is ~5 nT and the time scale was ~500 s. If we take a localized scale size of dy = 1000 km, it gives the elec-tric field intensity of 0.01 mV/m (=1000 x 10ˆ3 x 5 x 10ˆ-9 / 500). This value seems to be too small to cause the electron flux enhancement at L<1.2, because this value is two orders smaller than the prevailing electric field in the ionosphere by the thermo-spheric neutral wind through F-region dynamo. Thus, electric field associated with the observed magnetospheric compression seems not to work for the present case.

4. In Figure 2b, I noticed that not only the electron flux at L<1.2, but also the electron flux at high latitudes above +-60 degree increases, particularly at negative longitudes in the norther hemisphere and positive longitudes in the southern hemisphere. Thus the electron acceleration seems to be not confined at L<1.2. Why the authors neglect this

clear enhancement of electron flux about +-60 degrees? It is not clear whether the flux at middle latitudes increased or not in this color scale. If possible, it would be better to show the latitudinal profile of electron flux changes at some particular longitudes (e.g., at -120 degree) in a separated figure. Such figure may be useful to discuss how the electric field penetration suggested by the authors affect from high to low latitudes.

5. Sorbo et al. (GRL, 2006) indicated the >30keV electron flux enhancement in the NOAA/POES data at the equator caused by precipitation of energetic neutral atoms (ENAs). Although their events are mainly during magnetic storms, we can expect some amount of ENA flux even during quiet times, because ring current is a persistent feature in the magnetosphere. Is there any possibility that the present L<1.2 electron flux enhancement is related to the ENAs from the magnetosphere?

Sørbø, M., F. Søraas, K. Aarsnes, K. Oksavik, and D. S. Evans (2006), Latitude distribution of vertically precipitating energetic neutral atoms observed at low altitudes, Geophys. Res. Lett., 33, L06108, doi:10.1029/2005GL025240.

Technical Corrections

6. line 120-122: Please provide the values of the electric field suggested by these references.

7. line 182 (kept at the enhanced level for several hours): Readers cannot understand how the authors obtain the information "several hours" from Figure 1b. Please explain.

8. line 349-350 and line 462: I think we cannot exclude the possibility of solar wind dynamic pressure variations, since the OMNI solar wind dynamic pressure in Figure 3a shows small variations with time scales well less than 1 hour throughout the plotted interval.

9. lines 528-529: Why the authors focus only on night injections occurring occasionally from ~1300 to ~1700UT at 2-5 LT in Figure 2? There is a continuous injection at nearby 06 LT.

10. Figure 3: I cannot see shaded box at 13-23 UT, which is mentioned in the figure caption.

---

## Author Comment (AC1) · 4 Mar 2019

Dear Referee1,

Thank you very much for your comments and suggestions. We revised the Discussion and provided additional solid arguments and some quantitative estimation to support our suggestions. Here we try to address all your concerns.

General comments by Referee1:

"This manuscript reports a series of >30 keV electron flux enhancement events that happened at L<1.2 observed by POES satellites, and massive related observations

from THEMIS, ground magnetometer, ACE, etc. These events are likely to be a subset of the events analyzed in Suvorova (2017) and this study is a follow-up work related to Suvorova (2017). In the present study, the authors propose that the magnetic perturbation near the magnetopause causes a mixture of magnetosheath plasma and magnetospheric plasma to precipitate in high latitude (high L regions) which further induce a large transient electric field that could transport the electrons to L<1.2. However, there is no solid evidence reported to prove that the flux enhancements at L<1.2 are caused by magnetic perturbation near the magnetopause, nor analysis on the possibilities that this proposed chain of processes could work. The reviewer suggests to at least add in some more solid arguments or simulation results to prove that the proposed processes are reasonable before the paper can be published. The reviewer also suggests the authors to be more concise on some part of the paper, to avoid extra confusions of the readers."

Reply:

We thank the reviewer for the suggestion which help to improve the manuscript. We provide some additional observations and estimations in Discussion. We revised some descriptions in the paper in order to make them shorter.

In our study, we did not state or suggest that the magnetic perturbations cause the electron enhancements at low L-shells and plasma precipitations at high L-shells. Addressing to the comment 2, we will clarify this crucial point, which is important for overall understanding of our concept.

Specific Comment 1

(1.1) The authors presented the >30 keV electron flux measurements by POES satellites in Figure 1. In Figure 1, it is clear that electron fluxes are enhanced in the quasitrapped region (outside of SAA), but the fluxes in SAA that are more stably-trapped almost remain the same.

Reply:

Actually, electron fluxes did increase in the SAA region. However, the background fluxes in SAA were already high (several units of 105 (cm2 s sr)-1). The fluxes of FEE were mostly less than 105 (cm2 s sr)-1. Hence, they produce a little increase of the flux in SAA which is hard to be seen in the logarithmic scale. However, this effect is beyond the scope of our study.

(1.2) The authors refer to those events as injections in many places in the paper (e.g., line 202, 208). However, if those electrons are injected from higher L, they are supposed to become more 90 degree peaked in pitch angle, which means they are more likely to be stably-trapped and more enhancements in the SAA region are expected. From Figure 1, the slot region is not filled, which is supposed to be seen in an typical injection event that penetrates down to L=1.2. In fact, previous studies such as Li et al (2017, titled "Measurement of electrons from albedo neutron decay and neutron density in near-Earth space") reported events that enhanced stably trapped electrons are observed due to geomagnetic activities while the quasi-trapped electron fluxes stay the same. Moreover, people would easily link the enhancements in the quasi-trapped electrons to enhanced pitch angle scattering. The authors should show more detailed observations of these events and explain why these events are injections.

Reply:

We thank the reviewer for the comment. In order to clarify this crucial issue we revised Figure 2 (see below) in order to show the time profiles of intensities and L-shells of FEE enhancements. As we wrote, we use measurements from the vertically oriented detector ("0-detector") of electrons with local pitch angles of $90°$ at the equator (quasi or locally trapped electrons). Another detector measured precipitating electrons at the equator. The electron precipitations did not arise. The observed time profiles of quasi-trapped electrons are proper for injections.

We revised the text accordingly:

"Figure 2 and Table 1 present main characteristics of 15 FEE enhancements detected along equatorial passes of POES satellites (P2, P5, P6, P7, P8). We analyze the peak fluxes in the FEE enhancements (time, local time, longitude, and L-shell). " "As seen in Figure 2a,b, the FEE enhancements peak at minimal L-shells, i.e. at the equator. The fluxes decrease quickly with growing L. This pattern corresponds to a fast radial transport (injection) of electrons from the inner radiation belt. Note that pitch-angular scattering of electrons gives different profiles: the fluxes should be minimal and the equator and grow with L-shell."

Concerning to the albedo neutron mechanism. First of all, that paper is about relativistic electrons ($\sim$500 keV) during a geomagnetic storm. We think it has a little relation to our study of low-energy (>30 keV) electrons during nonstorm conditions. Next, it is impossible to apply this mechanism to the FEE enhancements because of the following well known facts: (1) The fluxes of albedo neutrons at equatorial latitudes are much lower (order of magnitudes) than the fluxes of FEE. (2) During magnetic quiet, the latitudinal profile of secondary particles (generated in decay) is positive, i.e. the flux of secondary particles increases with latitude due to a decrease of the cut-off rigidity of incident cosmic rays. Figure 2 demonstrates totally different pattern.

(1.3) The author should also specify the looking direction of the detector in the caption of Figure 1.

Reply:

In the caption of Figure 1 we add the following sentences: "The electrons are detected in vertical direction. In the forbidden zone, those electrons are quasi-trapped."

Specific Comment 2

(2.1) As is stated in the general comments, the authors have not present any solid evidence that the electron enhancements at L=1.2 could be caused by magnetic per-turbations near the magnetopause which is at quite large L. Only coincidences in time

are shown in the present study. The reviewer suggests to show more solid arguments or some simulation results to prove this possibility.

Reply:

We thank the reviewer for the comment, which help to improve the manuscript.

We want to clarify that we did not state that the observed magnetic perturbations near the magnetopause or inside the outer magnetosphere, at large L, can cause such a mixture of plasma at high latitudes and electron enhancements at the equator. These perturbations had small amplitudes of about of several to tens of nT [e.g., line 332]. We wrote [line 513] "A series of night injections of >30 keV electrons could be associated with transient magnetospheric magnetic field perturbations." We wrote about the association in other parts of text also. Note that the association is not a causality.

In Discussion we add:

"The amplitude of geomagnetic pulses is not very high: from few nT at ground to a few tens of nT at THEMIS. It should be noted that such magnetic perturbations are too weak to produce deep injections of >30 keV electrons below the radiation belt."

These magnetic perturbations are only a response of the geomagnetic field on occasional pressure pulses produced by magnetosheath plasma jets at the magnetopause. In the study, we emphasized an important role of transient subsolar foreshock condition, under which plasma jets are generated, for the magnetosphere–ionosphere coupling, particularly for non-storm events. The transient subsolar foreshock was only recently recognized as a major driver for a throat aurora at high latitudes, as we mentioned in the study.

In revised Discussion, we emphasize the importance of jets for the magnetosphere-ionosphere coupling under conditions of stable solar wind dynamic pressure and northward IMF:

"The interaction of jets with the magnetopause results in geomagnetic pulses and penetration of the magnetosheath plasma inside the magnetosphere (Figure 8). Note that the upstream conditions observed by THEMIS-C during both time intervals (from 12 to 16 UT and from 16 to 18 UT) were similar in that the quasi-radial IMF appeared. Hence, it is reasonable to suggest that the geomagnetic pulses occurred from 12 to 16 UT were also produced by jets because there were no strong enhancements in the solar wind dynamic pressure Pd. Indeed, as one can see in Figure 1 and 8, gradual tenuous variations of Pd do not exceed a few tenths of nPa and, thus, they cannot produce sharp geomagnetic pulses with amplitudes of ∼10 nT.

Also, in the revised manuscript, we provide additional arguments in favor to jet impact on the magnetosphere-ionosphere system. It was established that the impact results in magnetosheath particles precipitations at high latitudes near local noon. We present observations of hot plasma precipitations to the latitude ionosphere during the event:

[revised manuscript text omitted]

(2.2) In Li et al (2017), which is mentioned above, they also state that the large electric field can only cause an L shell distortion of 0.01 and this process is energy-dependent. Please comment on it and the possibility that the electric field moves the electrons to L<1.2 in this case.

Reply:

We note that Li et al. indicated the L-shell distortion of 0.01 for the relativistic electrons but not for the low-energy of 30 keV. In this concern, they cited the studies by Selesnik et al (2016) and Su et al. (2016), where observations of electron injections below L = 2 were compared with simulations. According to Su et al., "the electric field does not have a significant impact for electrons with energy >400 keV in the inner belt" (on page 8520), and "an enhanced large-scale electric field can be responsible for injection of ∼100 keV electrons in the inner radiation belt" (on page 8521). Also, it is important that they emphasized that "it is thus not necessary for electrons to be transported all the way from the outer zone during a single injection." Hence, as followed from this, the slot region is not necessary to be filled by enhancements (see Referee's comment 1). Selesnik et al. investigated various models of electric fields in applicability for deeper injections at L<1.2. Their conclusion is "Injection to L<1.2 is demonstrated in both observations and simulations by the end of 23 June, but the simulated injection is smaller because the model Ec (electric field) was reduced to zero for L<1.17".

The simulations showed that at L<1.3 during quiet condition, an average electric field is weak (∼0.4 mV/m), but for deep injections the field should be strong ∼5 mV/m. Hence, the simulation studies had to admit that strong electric fields could penetrate and cause deep injections at L<1.3, but mystery of a penetration mechanism was not disclosed.

We put attention that, in our "quiet" case the penetration electric field could not be generated in a storm/substorm process. What process could provoke such a strong electric field at L<1.3? The both studies pointed out that none of the existing models can accurately describe the penetration electric field and, hence, deep injections at L<1.2. Su et al. (2016) : "An accurate global electric field model is a necessary requirement in order to correctly capture the non-diffusive radial transport in the inner radiation belt." Our paper presents new experimental results, which help to develop a new model. The new model should be a subject of another study.

Specific Comment 3

(3) In Table 1, the authors list a series of flux enhancement events observed by POES. The author should specify the criteria used to select those events, and show some detailed electron flux profile of those events, such as how long the enhancements last, specific L shell of each event or how many data points are included in each event. The reviewer also suggests to use more commonly used names for POES satellites such as POES-15/18…instead of P2/P5…

Reply:

In the revised Figure 2, the intensity and L-shells of enhancements are shown. In the revised manuscript, we specify the criteria more precisely:

"… the forbidden zone extends at L < 1.2 in the latitudinal range from -20° to +30° and in the longitudinal range from 0° to 260°E (or 100°W) that is beyond the South Atlantic anomaly (SAA). … Figure 1b shows the interval 12 - 24 UT, when fluxes of >30 keV quasi-trapped electrons in the forbidden zone increased by 3 orders of magnitude above a background of ∼102 (cm2 s sr)-1 and kept at the enhanced level for several hours. We have selected FEE enhancements with intensity >103 (cm2 s sr)-1. "

We think that abbreviation P2, P5 etc. are more convenient for presentations in Figures and Tables. Moreover, we would keep the abbreviations, which we used in our previous papers.

Specific Comment 4

(4) Line 210-227: the authors intend to prove that each flux enhancement event is individual and not caused by any other event, for example, F2 is not caused by F1. However, this analysis is based on the presumption that the event is really transient. The authors should show some evidence to argue such as F1 could not have been enhanced 100 min before the observation of F1. Also, please explain why this is important. The reviewer does not find it very essential to the analysis later and suggests

to be more concise on this problem.

Reply:

In the original manuscript, we have already clarified this important issue: "Figure 1a shows the observations of the >30 keV electrons at 0 - 12 UT, before the enhancements occurred."

In the revised manuscript we provide an additional explanation:

"At that time, the satellites passed the same regions but they did not detect any FEE enhancements." The suggestion of multiple injections is important because several injections are accompanied by several jets. We find correspondence between the jets/pressure pulses and injections (Figure 11). Note that there were no substorm-associated injections in the present case.

Specific Comment 5

(5) Line 227: Please specify if these events are a subset of Suvorova (2017) event list. If so, the authors should make a clarification before stating that the characteristics agree with those in Suvorova (2017), otherwise it is misleading.

Reply:

This event (15 peaks in one day) is a subset (1%) of the total statistics of 2465 peaks at the equator within 530 days. Examples of storm and nonstorm enhancement events (including the interval of 1 - 3 August 2008) were presented in Figure 1 of the paper by Suvorova (2017). We mention it in Introduction of the revised manuscript:

"Note that this event is a subset (1%) of the total statistics collected by Suvorova (2017) during various conditions, from magnetic quite to extremely strong geomagnetic storms." Generally speaking, an individual event could be different from the overall statistics regarding the location of injections (local time and longitude ranges), especially for such specific conditions. Indeed, this particular event occurred under very

quiet geomagnetic condition, while the vast majority of events with similar parameters, such as multiple peaks or long durations of >4 h and high peak intensity > 104 - 105 (cm2 s sr)-1 occurred mainly during storms/substorms. Nevertheless, the electron enhancements during the August 1, 2008 event are in well agreement with those found from statistics as we concluded in the original manuscript (lines 223 – 227), namely: "specific longitudinal and local time distributions of the enhancements indicate multiple injections during about 4.5 h in the sector of 0 - 6 LT, and the injection region was confined within 3 h of local time over central and eastern Pacific. In general, these characteristic of injections are in well agreement with those found from statistics (Suvorova, 2017)."

Specific Comment 6

(6) Figure 3: Since the authors show that L1 is not a preferable location for observations of the magnetic perturbations as compared to Themis, this figure is not necessary. The reviewer suggests to combine some of the information in Figure 3 into Figure 4 and be more concise on the text as well, in order to help the readers to focus on the important part, Themis observations.

Reply:

The Figure 3 is important at least because of the comment by Referee 2, who believes that variation in the solar wind dynamic pressure could cause these magnetospheric compressions. The Figure clearly demonstrates that variations are tenuous. Additionally, a discussion of the OMNI data is important due to its wide use.

Specific Comment 7

(7) Line 611: Please use explicit number of the latitude of throat aurora instead of "lower latitude" here. It is misleading because this study is talking about phenomena at L=1.2 (<30 deg in latitude), while the throat aurora in a series of Han et al papers is still located at >70 deg in latitude (or please correct this number).

Reply:

Thank you for the important comment. We revised the text accordingly:

"Sometimes, the dayside aurora penetrates to lower geomagnetic latitudes of $\sim72°$ from the discrete aurora oval at geomagnetic latitude $\sim76°$, so-called throat aurora."

Sincerely, Alla Suvorova

Please also note the supplement to this comment:
https://www.ann-geophys-discuss.net/angeo-2019-5/angeo-2019-5-AC1-supplement.pdf

**Fig. 1.** Figure 2

[Figure]

**Fig. 2.** Figure 11

---

## Author Comment (AC2) · 12 Mar 2019

Dear Referee2,

Thank you very much for your comments and suggestions. We revised the Discussion and provided additional solid arguments and some quantitative estimations to support our suggestions.

General Comments by Referee2:

This paper reports energetic (>30 keV) electron flux enhancement at L<1.2 measured by the NOAA/POES satellites and relate it to the transient injection of magnetosheath

plasma into the dayside magnetopause region, which is measured by the THEMIS satellite, and global geomagnetic pulses, which are measured by ground INTERMAG-NET magnetometers and GOES satellites. The authors propose a scenario of possible association between these dayside magnetopause phenomena with the deep injection of >30keV electrons at L<1.2 by the penetration of localized electric field. The electron flux enhancement at L<1.2 is well described including its research history which is very interesting. Looking through this paper, however, I think the connection between the observed phenomena occurring in the dayside magnetosheath/magnetopause region and the electron flux enhancement at L<1.2 is weak and not well validated by the observations reported in this paper. These two phenomena occur in the same half day of 12-24 UT on August 1, 2008. But there is a significant possibility that they occur in the same day "by chance". I think it is necessary to provide some more concrete evidence including some quantitative estimation that can explain the observed L<1.2 electron enhancement.

Reply:

We thank the reviewer for the suggestion which help to improve the manuscript. In the revised manuscript, we add some estimation on ExB drift. We provide additional arguments in favor to jet impact on the magnetosphere-ionosphere system. We present observations of hot plasma precipitations to the high-latitude ionosphere during the event in a new Figure 11. It was established that the jet impact results in magnetosheath particles precipitations at high latitudes near local noon.

In Discussion we add:

[revised manuscript text omitted]

Specific Comment 1

1. The descriptions of OMTI, THEMIS, GOES and ground magnetometers are fair and easy to understand, although they can be shorter. The authors propose a scenario that dayside magnetopause phenomena cause magnetosphere compression, and associated magnetosheath / magnetospheric plasma precipitation to the dayside ionosphere at high latitudes that result in a local increase of the ionospheric conductivity. This in turn promotes generation of transient localized electric fields, which are able to penetrate from high latitudes to very low latitudes to accelerate energetic electrons at L<1.2. However, in the nightside auroral zone we have normal aurora and associated ionospheric conductivity change which can be much larger than those in the dayside aurora. If the scenario proposed by the authors works, why we do not have L<1.2 acceleration during ordinary (non-storm time) substorms which occur almost every day and cause strong aurora and associated conductivity change in the nightside high latitudes? If L<1.2 electron flux enhancement does not occur during ordinary substorms, I think it indicates that the proposed scenario does not work in the actual magnetosphere.

Reply:

Indeed, a typical substorm produces an increase of conductivity on the nightside. In

contrast, our scenario is proposed for the magnetic quiet (no substorm!) and it is based on the change of dayside conductivity, which should be larger then the nightside one. The scenario explains qualitatively the induction of electric field on the nightside.

It is true that not every substorm results in FEE enhancement. It means that substorm activity alone is insufficient for induction and penetration of electric field to low latitudes on the nightside. We also pointed out this important issue in Introduction of original manuscript [lines 137-146]. Namely, the FEE events account for only 8% of the total time from 1998 to 2016 (Suvorova, 2017). Most of FEE events are accompanied by substorm activity, but we have found "three dozen days without essential substorm activity". Hence, it should be something else. Here, we totally agree with the Reviewer. It seems the factor controlling the occurrence of FEE enhancements might be different for storm-time and non-storm conditions. In the previous study, we have found that the illumination of the dayside auroral zone plays the key role, because its dependence on tilt angle explains perfectly the annual variation of FEE occurrence with a main maximum during the northern summer period, from May to September (see Figure 13 in Suvorova, 2017).

In order to clarify this important issue, we have revised the end of Introduction accordingly:

"External drivers from the solar wind should trigger some processes in the magnetosphere-ionosphere system that might result in the electron injections into the forbidden zone. However, the external drivers are necessary but often not sufficient for FEE enhancements to occur. If the auroral ionosphere is sunlit, then impact of external drivers more likely results in the electron injections into the forbidden zone. In this case, the factor of the dayside auroral ionosphere conductivity is sufficient, and it comes to the fore during weak geomagnetic activity. The relevant processes in the magnetosphere-ionosphere chain during magnetic quiet are still unclear."

In order to clarify the role of the dayside conductivity in the auroral zone, we add Figure
11 (see above). As known, the initial response to the solar wind impact is particle precipitations to the high-latitude ionosphere at the dayside, particularly within the cusp region. For example, impact of high solar wind pressure under northern IMF Bz. It is a very common case to observe dayside aurora and intense particle precipitations in the cusp. Under non-substorm condition, intense dayside particle precipitations in sunlit auroral zone can provide a temporal condition for a higher conductivity at the dayside. In such condition, the electric field is induced in the nightside ionosphere (where the conductivity is relatively lower) and then the induced electric field might penetrate to low latitudes providing the earthward transport of particles. Hence the additional ionization of sunlit auroral zone is a very important condition in the proposed scenario. We have to remind that the mechanism of the electric field penetration is still unresolved problem of the magnetospheric physics.

Specific Comment 2

2. As shown in Figures 7 and 8 the THEMIS satellites shows repeating motion in and out from the magnetosphere to the magnetosheath. Such in and out features are very often seen when THEMIS is approaching to the magnetopause region, because the magnetopause location is not fixed and changes due to dynamic pressure change in the magnetosheath and/or surface waves caused by Kelvin-Helmholz instability in the magnetopause. In the present case, since compressional wave signatures are seen in GOES and ground magnetometers, it is likely that the dynamic pressure variation outside the magnetosphere is the cause of this motion of THEMIS in/out from the magnetosphere. But I think such compressional wave with an amplitude of a few nano-tesla is not unusual and occur frequently. Then how often does the authors find L<1.2 electron acceleration? Is this a frequent phenomenon occurring associated with the frequently-occurring compression of the magnetosphere with the amplitude of a few nano-tesla on the ground magnetometers? How the authors can prove that these two phenomena occurs in the same time not by chance? Maybe the authors can check correspondence of timing between each magnetospheric compression and the electron flux enhancement at L<1.2.

Reply:

We have partially replied to this comment above (see Reply to General comment). Concerning to the magnetopause motion and magnetic variations in our case, the weak magnetic pulses do not affect the FEE enhancements. They are just signatures of jets impacting the magnetopause. In the scenario proposed, the key effect is the jet-related penetration of the magnetosheath plasma inside the magnetosphere and its precipitation to the dayside auroral ionosphere. Actually, only a small portion of jets (∼10%) pierces the magnetopause. In the revised manuscript, we mention: "On the other hand, when a jet hits the magnetopause, the magnetosheath plasma is not necessarily penetrating into the dayside magnetosphere and, hence, precipitating at high latitudes [Dmitriev and Suvorova, 2015]."

This is why not every magnetic pulse is followed by FEE enhancement. Considering Figure 11 in the revised manuscript, we demonstrate the direct relationship between jets, magnetosheath plasma penetration/precipitation and FEE injections.

Specific Comment 3

3. The authors show magnetic field pulses observed by GOES and ground magnetometers. If the penetrating electric field is propagating in the magnetosphere, it should be related to the observed magnetic field variations by the Maxwell's equation of dB/dt = -rot E. One can argue that the observed magnetic field variation (dB/dt) can be used to estimate electric field by taking only one component of the rotation, e.g., dB/dt = dEx/dy (dEx = dB/dt * dy). The GOES magnetic field amplitude is _5 nT and the time scale was _500 s. If we take a localized scale size of dy = 1000 km, it gives the electric field intensity of 0.01 mV/m (=1000 x 10Ȩ̈3 x 5 x 10Ȩ̈-9 / 500). This value seems to be too small to cause the electron flux enhancement at L<1.2, because this value is two orders smaller than the prevailing electric field in the ionosphere by the thermospheric neutral wind through F-region dynamo. Thus, electric field associated with the
observed magnetospheric compression seems not to work for the present case.

Reply:

We agree that the magnetic filed pulses were too weak to provide the FEE injections. To make the text clear, in Discussion we add:

"The amplitude of geomagnetic pulses is not very high: from few nT at ground to a few tens of nT at THEMIS. It should be noted that such magnetic perturbations are too weak to produce deep injections of >30 keV electrons below the radiation belt."

Specific Comment 4

4. In Figure 2b, I noticed that not only the electron flux at L<1.2, but also the electron flux at high latitudes above +-60 degree increases, particularly at negative longitudes in the norther hemisphere and positive longitudes in the southern hemisphere. Thus the electron acceleration seems to be not confined at L<1.2. Why the authors neglect this clear enhancement of electron flux about +-60 degrees? It is not clear whether the flux at middle latitudes increased or not in this color scale. If possible, it would be better to show the latitudinal profile of electron flux changes at some particular longitudes (e.g., at -120 degree) in a separated figure. Such figure may be useful to discuss how the electric field penetration suggested by the authors affect from high to low latitudes.

Reply:

We thank the Referee for the valuable comment. Concerning the mechanisms of radial transport, we wrote in Introduction [lines 55-61] that studies (e.g., Turner, 2015) showed that mechanisms of injections and dynamics of energetic electrons at low L-shells (inside the plasmasphere, L<4) are different from those at higher L-shells (outside the plasmasphere). Nevertheless, we have checked > 30 keV electron fluxes at high latitudes using data from both detectors measured precipitating and trapped populations of the outer radiation belt. We found that the fluxes of the radiation belt electrons increased after 16 UT. Note that increases of electron fluxes at high latitudes observed

at low-earth's orbits are rather caused by pitch-angular scattering of trapped electrons into the loss cone due to wave activity rather than due to the effect of electric field.

From 12 to 16 UT the electron fluxes at high latitudes were not disturbed in contrast to FEE at low latitudes. Note that at middle latitudes, the satellites measured the background intensity of precipitating electrons from the inner radiation belt, while trapped population was observed in the SAA region (L<2). As seen in Figure 1 (a, b), the background fluxes in SAA were already high (several units of 105 (cm2 s sr)-1), and the fluxes of FEE were mostly less than 105 (cm2 s sr)-1. Hence, they produce a little increase in the flux at L<2.

At higher L-shells, the increase could be even less. Let assume that the induced electric field accelerates electrons in the radiation belt. In this case, the electrons should stay in the acceleration region for a certain time, which should be sufficient for effective acceleration. In Discussion we consider this situation for >100 keV electrons in the inner radiation belt:

"In contrast, the >100 keV electrons with the azimuthal period of ~6 h leave quickly the injection region and, thus, do not have enough time to penetrate to the forbidden zone. This effect can explain the absence of high-energy electrons in the FEE enhancements presented."

Similarly at high latitudes, i.e. for the outer radiation belt (L-shells ~ 4), we can find that ~30 keV electrons have azimuthal period of ~6 h and, thus, they leave quickly the acceleration region and gain not too much energy. So the flux increase (if any) is hard to be seen in the logarithmic scale. In the original manuscript we have already discussed how the electric field penetration affects from high to low latitudes. Namely, penetration of electric field is still a serious problem, and modern models can not provide strong electric field at L<1.3 in order to explain observation of deep injections (e.g., Su et al., 2016; Selesnick et al., 2016). In Introduction and Discussion [lines 122-128; 647-652] we emphasized that:

"and most of researchers consider and model an electric drift of electrons in the ExB fields, even though the electric field must be very high (e.g., Zhao and Li, 2013; Turner et al., 2015; Lejosne and Mozer, 2016; Selesnick et al., 2016; Su et al., 2016; Zhao et al., 2017a). There is no explanation for penetration of a strong electric field to such low L-shells. ...... empirical electric field models are limited and do not provide the results below L~2 (e.g., Rowland and Wygant, 1998; Matsui et al., 2013). The most modern research suggests that the actual strength of penetration electric fields can be stronger than any existing electric field model at L < 2 (Su et al., 2016)."

"Another serious problem is the generation/penetration of electric fields in the inner magnetosphere, which is far from complete understanding. Numerical estimations show that the anomalous (fast) radial transport of particles observed in the inner magnetosphere can be produced by the electric field up to 5 mV/m (Selesnick et al., 2016; Suvorova et al., 2013). At the present time, there are no models predicting strong electric fields in the inner radiation belt and below."

Specific Comment 5

5. Sorbo et al. (GRL, 2006) indicated the >30keV electron flux enhancement in the NOAA/POES data at the equator caused by precipitation of energetic neutral atoms (ENAs). Although their events are mainly during magnetic storms, we can expect some amount of ENA flux even during quiet times, because ring current is a persistent feature in the magnetosphere. Is there any possibility that the present L<1.2 electron flux enhancement is related to the ENAs from the magnetosphere? Sørbø, M., F. Søraas, K. Aarsnes, K. Oksavik, and D. S. Evans (2006), Latitude distribution of vertically precipitating energetic neutral atoms observed at low altitudes, Geophys. Res. Lett., 33, L06108, doi:10.1029/2005GL025240.

Reply:

The ENA mechanism cannot explain very strong (3 order of magnitude) enhancements of the count rate in the channel of >30 keV electrons, which was observed during

magnetic quiet (!) in a wide range of latitudes (∼40 deg) and in a restricted range in longitudes. In particular, it is impossible to explain why almost whole Eastern hemisphere (longitudes from 0 to 160E) is free from ENA. We even do not see there a small-amplitude equatorial maximum of ENA from the quiet ring current. Hence, this is certainly not the ring current effect.

Technical Corrections

6. line 120-122: Please provide the values of the electric field suggested by these references.

Reply:

We add the following : "According to simulation results of Selesnick et al. (2016), the electric field of ∼5 mV/m can provide deep injections at L<1.3."

7. line 182 (kept at the enhanced level for several hours): Readers cannot understand how the authors obtain the information "several hours" from Figure 1b. Please explain.

Reply:

We add this information in the next paragraph: "Figure 2 and Table 1 present main characteristics of 15 FEE enhancements detected along equatorial passes of POES satellites (P2, P5, P6, P7, P8). The fluxes kept at the enhanced level for several hours. We analyze the peak fluxes in the FEE enhancements (time, local time, longitude, and L-shell)."

8. line 349-350 and line 462: I think we cannot exclude the possibility of solar wind dynamic pressure variations, since the OMNI solar wind dynamic pressure in Figure 3a shows small variations with time scales well less than 1 hour throughout the plotted interval.

Reply:

The OMNI data are obtained from measurements by the ACE and Wind upstream

monitors. The ACE data were shown in Figure 8. The both datasets showed tenuous solar wind pressure variations of a few tenths of nPa around an average value 1.2 - 1.8. Figure 8 (b1, b2, b3) allow estimating the pressure variations more accurately (< 0.2 nPa). They cannot produce magnetic pulses with amplitude of ∼10 nT. We check the both data further and find fast ∼10% fluctuations of pressure with a quasi-period varying from ∼2 to 10 min. The timescale of the magnetic pulses is much longer, from 15 min to ∼1-1.5 h (Table 2). Hence, it is hard to connect fast pressure variations with the occasional magnetic pulses and shallow valleys presented in Figure 6. Moreover, small-scale fluctuations of the solar wind pressure could not produce intense precipitations observed in the cusp region (see Figure 11).

In the revised manuscript we add:

"Indeed, as one can see in Figures 1 and 8, tenuous variations of Pd do not exceed a few tenths of nPa and, thus, they cannot produce sharp geomagnetic pulses with amplitudes of ∼10 nT."

9. lines 528-529: Why the authors focus only on night injections occurring occasionally from ∼1300 to ∼1700UT at 2-5 LT in Figure 2? There is a continuous injection at nearby 06 LT.

Reply:

We replace "night" to "nightside" to avoid misunderstanding. Actually the "continuous injection at nearby 06 LT" occurred at around 5 LT (see Table 1). We analyzed 8 peak fluxes on the nightside at 1.8, 1.6, 5.1, 5.0 and 4.9 LT. They were listed as F1-F6 and F8, F10 in Table 1.

We revised the text accordingly:

"With analysis of longitudinal and local time distributions of the enhancements we identified a series of nightside injections occurred in the sector of 2 - 5 LT during the period from ∼1300 to ∼1700 UT (Figure 2)."

10. Figure 3: I cannot see shaded box at 13-23 UT, which is mentioned in the figure caption.

Reply:

We correct the Figure 3.

Figure 11. Dynamics of the geomagnetic field and particles on 1 August 2008: (a) FEE enhancements, (b) plasma precipitation at high latitudes, and dayside magnetic field perturbations observed by (c) GOES-12 and (d) THEMIS-D. The numbers indicate the FEE injections at ∼2 and ∼5 LT (see Table 1), colors for POES satellite are the same as in Figure 2. Plasma precipitations are shown for the energy flux above the threshold of 0.5 (erg/sm2 s) and are grouped in LT: 23 – 24 LT (light gray), 0 – 2 LT (gray), 5 – 6 LT (blue), 12.5 - 15 LT (red points), 15 – 16 LT (violet), and 19.5 – 21.5 LT (green).

Sincerely,

Alla Suvorova
* * *
**Fig. 1.** Figure 11

---

## Referee Report (RR1)

This manuscript reports on an event from 1 Aug 2008 where >30 keV electron flux is observed to be enhanced in the drift-loss cone ("quasi-trapped" electrons) at very low L values, L<1.2. Electron flux is rarely observed to be elevated in this region and understanding the physical processes that can transport energetic electrons to such low L values is interesting and worthy of study. Several recent works have been looking at what controls the energetic electron dynamics in these inner regions, so this topic is timely. The injection event is noteworthy also because it occurs during very quiet conditions; the only activity present is a weak, isolated substorm of (~300 nT) that occurred roughly 4 hours after the injection activity begins. Such deep energetic particle injections are usually associated with intense substorm activity but this event is not, and thus presents a unique challenge to understand what causes the deep injections. The authors note that they have found more than three dozen days (over a 10 year period) with such 10s keV electron injections but without substorm activity. Previous work by the author (Suvorova (2017)) has suggested an important role of the auroral ionosphere in the occurrence of these injections.

The present work builds on these ideas by presenting a scenario where transient foreshock waves and magnetosheath plasma jets cause global perturbations in the geomagnetic field. These compressions introduce magnetosheath plasma into the magnetosphere, which then precipitates to the dayside ionosphere at high latitudes. This in turn results in a local increase of the ionospheric conductivity and generation of transient localized electric fields, which are able to penetrate from high latitudes to very low latitudes. The authors estimate that the strength of these electric fields would need to be ~5 mV/m to produce the deep injections, which is consistent with prior estimates. This whole proposed scenario is supported only by circumstantial evidence, but it is at the very least plausible. In that sense, I agree with the other referees in their assessments that the causal links are weak and could be just due to chance. However, whether or not the scenario is viable, the observations themselves are unique and worthy of publication in the scientific literature. Perhaps the scenario could be softened as merely a speculative after thought, and the paper changed to more focus solely on the observations? That way an educated reader can judge for themselves whether or not the scenario presented is indeed viable. I also have some concerns related to the way the data is analyzed and presented, particularly with regards to the wave amplitudes and peaks identified, that I would like to see addressed before I can recommend the manuscript for publication.

**Major Comments**

**1. Several aspects of the analysis are somewhat qualitative and should be made more quantitative.**
For example, at L351: *"Prominent magnetic peaks are indicated by dashed lines and listed in Table 2."* – what is the criteria for determining these peaks? It is not stated, and there are clearly some peaks in the same interval that are not identified or called out as peaks. (e.g., at 1445 UT, 1350 UT…). Why are these peaks not included? I suggest that the authors use a quantitative criterion to identify the peaks, so that they are not ambiguously and arbitrarily chosen. It almost seems as though they are chosen to match the ground magnetic perturbations shown in Figure 9. I also suggest that the authors detrend the three THEMIS time series and the ground magnetometer data so as to reveal the peaks in all the time series more clearly. Note that this will help better confirm the claim on L421 that "*the first magnetic pulse at ~1200 UT can not be emerged from THEMIS data because of the large background magnetic field in the inner magnetosphere.*" Background trends can easily (and should) be removed by detrending.

Similarly, at L497 the authors state: "*Smaller amplitude at daytime is a result of an amplifying integral effect from the Chapman-Ferraro current at the magnetopause and ionospheric Sq-current at the ground.*" - The ground magnetometer data are not presented in such a way that one can determine whether or not the ground field perturbations are weaker on the dayside than the nightside. The data need to be detrended. The scales are larger on the dayside stations (going up to >10 nT in some cases) so it is difficult to determine from the figure whether these perturbations are lower than those on the night side stations. I suggest that the data be detrended and Fourier analyzed to calculate the RMS wave power at each station to quantitatively assess the amplitude of the ground perturbations for comparisons.

Also, at L567, the authors state: "*We find that the magnetospheric ULF waves are not strong enough to produce anomalous radial transport of energetic electrons at L < 1.2*" – how is this determined? I see no discussion along these lines or any such calculations anywhere in the manuscript.

Finally, it is noted that the first magnetometer pulse is at 1330 UTC, which is after the first appearance of >30 keV electrons observed at L<1.2. This is not entirely consistent with causality, with the perturbations leading to enhanced electric fields that produce the injections. How do the authors reconcile this?

**2. The role of dynamic pressure variations:** At L363: "*The magnetic variations associated with compression-expansion effects could not be caused by the solar wind pressure variations, which were gradual and small during the interval (see Figure 3).*" Here the authors are referring to the OMNI data as evidence for this claim, which is supported by the OMNI data. However, the authors have just argued that there are significant differences between the OMNI data and what is actually observed just upstream of the magnetosphere by TH-C. Thus, it seems as though TH-C data should be presented, in terms of in-situ pressure variation observations. The authors go on to say that THEMIS cannot observe in the magnetosheath at this time, but what about what TH-C observes locally in the solar wind just upstream of the bow shock? Are there pressure variations (magnetic or dynamic) observed there? I would like to see those data, as they would bolster these claims significantly. In addition, at L477 the authors state: "*we expect that variations in the geomagnetic field (if any) should result from the local magnetosheath pressure pulses.*" Why? There are a number of other mechanisms that can cause activity in ground magnetometer during relatively quiet times (e.g., ULF waves driven by Kelvin Helmholtz; ULF waves generated internally by plasma instabilities, etc…). Have the authors considered any of these? Why do they believe that these are not occurring at the time of the ground perturbations?

**Minor Comments**

- Figure 3 caption: "The shaded box denotes the time interval from 13 to 23 UT" – there is no shaded box in the figure
- L171: "quite" -> "quiet"
- L181-182: You might mention here that this is the "0-degree telescope," since this is how it is commonly referred to in the literature.
- L183: Is this the definition of the forbidden zone? If so, you should state that "The forbidden zone is defined as L<1.2 ….." What field model are you using to define the L values?
- L193-194: "*Fluxes of the >100 keV electrons and >30 keV protons did not increase also (not shown).*" You should indicate here whether you are referring to quasi-trapped, precipitating, or both.

- L199: This labeling of the POES vehicles is not standard. Is this what you mean? "(P2 = MetOp2, P5 = NOAA-15, P6 = NOAA-16, P7 = NOAA-17, P8 = NOAA-18)"? If so, you should state this here.
- L220-221: "*All remaining enhancements F2, F3, F5, F6, F8 and F10 of >30 keV electron fluxes were observed in the early morning (5 LT) for a long time interval of ~4 h*" – I don't know how you can easily see this from the figure. I think you need to label each of the curves in Figure 2(a) with the corresponding FEE number.
- L283: How do you know that TH-C is upstream of the bow shock? There's no bow shock model shown. Is this simply inferred from the TH-C measurements?
- L287: "GMS" -> "GSM"
- L300: "After 1500 UT, the OMNI data do not match the TH-C observation any more, even with time correction." – it would be nice if you showed also a smoothed version of the TH-C cone angle time series in Figure 5(c )
- L401: Are the ACE data time shifted here?

---

## Referee Report (RR2)

The observations presented in this paper that show the appearance of >30 keV electrons in the "forbidden" region during quiet times is very interesting and worthy of publications. I thank the authors for attempting to respond to my earlier criticisms, but I remained unconvinced of their claims relating the low L injections to dayside magnetosheath jet activity, which, through a complicated chain of events, is supposed to enhance the low latitude electric field on the nightside. I attempt to summarize my objections as follows.

As I see it, these facts are supported by the evidence presented manuscript, all of which I agree with:

- >30 keV electrons were observed at very low L, L<1.2 during a quiet interval
- They were likely injected from the nightside in the 2-5 MLT region
- They were not associated with substorm injection/activity
- There were global magnetic field perturbations observed throughout the dayside magnetosphere (GOES, THEMIS, ground mags) around the same time
- These global field perturbations were likely related to upstream foreshock activity/waves
- The field perturbations were too weak to produce radial transport of >30 keV electrons and were not the cause of the low L injections
- Foreshock pulses and associated magnetosheath jets were observed on the dayside

These represent very interesting and intriguing observations, particularly the appearance of electrons at very low L during quiet time. However, it is then argued that the magnetosheath jets cause hot plasma (50 eV - 10 keV) to precipitate into the dayside auroral region (L = 7 - 15) and that the jet-related magnetosheath plasma can produce significant additional ionization and increase conductivity of the high-latitude ionosphere on the dayside. It is then argued that this enhanced dayside conductivity enhances dayside currents in the ionosphere which "should in turn promote generation of transient localized electric fields on the nightside and especially in the postmidnight sector, where the conductivity is weak." I do not follow this logic and there are no additional arguments/calculations/references to support these claims. It is then hypothesized that "the induced nightside electric field might penetrate from high to low latitudes (very low L shells) and results in ExB drift of electrons to lower L-shells." I do not understand the mechanism that would allow this localized nightside electric field to penetrate from high to low latitudes. Again, there are no additional arguments/ calculations /references to support these claims. It is then argued that it is this electric field that produces the electron injections at very low L. In summary, I find these final arguments regarding the last chain in the (complicated) proposed scenario to be weak and unconvincing.

I will also comment that I see no relationship in Figure 11 between the magnetic field perturbations and the NOAA/POES/TED precipitation signatures. It is not demonstrated whether these TED precipitation signatures are exceptional or the norm. What do the TED measurements show before and after this interval?  When the magnetic field is quiet, are these plasma precipitations observed? I suspect that the TED measurements always look like this, but one cannot be sure from the manuscript. If they are, then that begs the question why does the mechanism proposed by the authors only occur in this event, and not all of the time? What is so unique about the magnetospheric state and the observations that allow access of >30 keV electrons down to very low L values in this rare event?

In summary, while I think that the appearance of electrons at very low L during quiet time is a very interesting question, the authors have not convinced me that their proposed scenario is plausible, and thus I cannot recommend this article for publication.

---

## Author Response (AR2)

Dear Referee1,

Thank you very much for reviewing the manuscript.
Text of the revised paper was shortened by 4.5 pages.

Dear Referee3,

Thank you very much for your constructive comments and helpful suggestions. We add data which demonstrate important role of dynamic pressure variations in the subsolar foreshock region. Correspondingly, sections 2 and 3 (Observations and Discussion) were substantially revised. We try to soften all formulations concerning the assumed scenario. The revised text is marked in blue. Here we address all your concerns.

*Major comments by Referee3:*

**1. Several aspects of the analysis are somewhat qualitative and should be made more quantitative.**
For example, at L351: *"Prominent magnetic peaks are indicated by dashed lines and listed in Table 2."* – what is the criteria for determining these peaks? It is not stated, and there are clearly some peaks in the same interval that are not identified or called out as peaks. (e.g., at 1445 UT, 1350 UT…). Why are these peaks not included? I suggest that the authors use a quantitative criterion to identify the peaks, so that they are not ambiguously and arbitrarily chosen. It almost seems as though they are chosen to match the ground magnetic perturbations shown in Figure 9.

Reply:
In selection of peaks, we paid more attention to a sequence "depletion-compression" in the magnetic field, because (and Reviewer is absolutely right here) these signatures are also found in the ground observations. Small-amplitude peaks noted by Reviewer can be considered as just subsidiary because they do not appear in Figures 9-10.
We clarify our selection of magnetic peak and change the text:
"Prominent magnetic "dimple-hump" structures are indicated by dashed lines (as 1, 2, and 3) and their peaks are listed in Table 2. We select peak-to-peak amplitudes exceeded ~5 nT in the GOES data (Figure 6c). The dimple-hump structures show the largest amplitudes up to 15 nT in THEMIS data (Figure 6b)."

Referee:
I also suggest that the authors detrend the three THEMIS time series and the ground magnetometer data so as to reveal the peaks in all the time series more clearly. Note that this will help better confirm the claim on L421 that *"the first magnetic peak at ~1200 UT can not be emerged from THEMIS data because of the large background magnetic field in the inner magnetosphere."* Background trends can easily (and should) be removed by detrending.

Reply:
We thank Reviewer for the suggestion. We detrend the THEMIS magnetic data and add or replace corresponding panels in Figures 6, 9, 10 and 11. Indeed, after this procedure a magnetic peak at 12 UT from the THEMIS-D data became more prominent.

It is quite difficult to detrend the ground magnetometer data, especially at daytime, because of long (5h) time interval. Also, our notice about amplitudes at the dayside ground stations is not important and has been removed.

We add the following text around Figure 6:

"The THEMIS magnetic data were detrended using the Tsyganenko T04 geomagnetic field model (Tsyganenko and Sitnov, 2005) and IGRF-2005 model (see Figure 6b). The IGRF model describes the Earth's main magnetic field and the T04 model represents magnetic fields from magnetospheric current system. "

Referee:
Similarly, at L497 the authors state: "*Smaller amplitude at daytime is a result of an amplifying integral effect from the Chapman-Ferraro current at the magnetopause and ionospheric Sq-current at the ground.*" - The ground magnetometer data are not presented in such a way that one can determine whether or not the ground field perturbations are weaker on the dayside than the nightside. The data need to be detrended. The scales are larger on the dayside stations (going up to >10 nT in some cases) so it is difficult to determine from the figure whether these perturbations are lower than those on the night side stations. I suggest that the data be detrended and Fourier analyzed to calculate the RMS wave power at each station to quantitatively assess the amplitude of the ground perturbations for comparisons.

Reply:
We thank Reviewer for this comment. We delete the paragraph including this sentence, because it is not significant in our study and we try to shorten the paper according the recommendation of another referee.

Referee:
Also, at L567, the authors state: "*We find that the magnetospheric ULF waves are not strong enough to produce anomalous radial transport of energetic electrons at L < 1.2*" – how is this determined? I see no discussion along these lines or any such calculations anywhere in the manuscript.

Reply:
The mechanism of fast transport with ULF-waves with amplitude of a few nT during non-storm condition was shown to be invalid for filling the slot region (L<3) by Park et al. (2010). We believe this conclusion undoubtedly is right for lower L (L<1.2). However, for shortening the paper, we delete everything about ULF waves from section 2.5 and as well as this statement (Line 567).

Referee:
Finally, it is noted that the first magnetometer pulse is at 1330 UTC, which is after the first appearance of >30 keV electrons observed at L<1.2. This is not entirely consistent with causality, with the perturbations leading to enhanced electric fields that produce the injections. How do the authors reconcile this?

Reply:
We already discussed this issue in details and presented Figure 11 (see Lines 536-539, 580-586 in Discussion of the revised manuscript). Particularly, we wrote:

(Line 628, in revised version Line 536) "The first FEE injection (F1) at ~1250 UT was preceded by several geomagnetic pulses observed by GOES-12…."

(Line 648; in revised version Line 570) "Previously, simulations by Su et al. (2016) have showed that it is not necessary for electrons to be transported earthward all the way during a single injection. Hence, we can consider a multi-step radial transport produced by a number of short pulses of $E$."

and

(Line 658, in revised version L 580) "…the first FEE injection requires a long time (~hour and longer) and several pulses of $E$ in order to transport energetic electrons from undisturbed edge of the inner radiation belt to $L$~1.1. Then, >30 keV electrons populate $L$-shells from 1.15 to 1.1 that makes possible to transport electrons to 900 km heights for a short time of ~10 min by one pulse of strong $E$. The latter pattern is applicable for the FEE injection F2 and others."

Here we shortly note that first injections (F1 at 1250 and F2 at 1315 UT) are weaker than subsequent ones and are preceded by several weak magnetic peaks. We add in section 2.4 (Line 322-323) : "From 11 to 13 UT, one can see several increases of a few nT observed by GOES and/or THEMIS at ~1125, ~1200, ~1245 and ~1300 UT (Figure 6b)."

**2. The role of dynamic pressure variations:** At L363: *"The magnetic variations associated with compression-expansion effects could not be caused by the solar wind pressure variations, which were gradual and small during the interval (see Figure 3)."* Here the authors are referring to the OMNI data as evidence for this claim, which is supported by the OMNI data. However, the authors have just argued that there are significant differences between the OMNI data and what is actually observed just upstream of the magnetosphere by TH-C. Thus, it seems as though TH-C data should be presented, in terms of in-situ pressure variation observations. The authors go on to say that THEMIS cannot observe in the magnetosheath at this time, but what about what TH-C observes locally in the solar wind just upstream of the bow shock? Are there pressure variations (magnetic or dynamic) observed there? I would like to see those data, as they would bolster these claims significantly.

Reply:

Thank you for the suggestion. In contrast to the OMNI pressure, TH-C observations show large variation of the dynamic (and total) pressure in the subsolar foreshock region.

We revised Figure 5 and Figure 7 adding panels with solar wind pressures from TH-C. In Table 2, we add a column with timing of the foreshock pressure pulses.

We add the following (Line 285-297):

"Figure 5d demonstrates large difference in solar wind dynamic pressure acquired from the TH-C probe, the ACE upstream monitor and OMNI data. The ACE data are shifted by 60 min. In contrast to OMNI and ACE, TH-C observed strong fast fluctuations in the dynamic pressure during intervals of subsolar foreshock (see Figure 5c). Note that ACE shows in average a smaller pressure than OMNI predicts, and it is more close to the TH-C observations. The fluctuations in the TH-C measurements are characterized by pressure pulses, which exceed sometimes the dynamic pressure from ACE (e.g., at 1320-1330, 1350, 1420, 1440, 1530 and etc.). The pulses were originated from plasma density enhancements because the plasma velocity remained practically constant at that time (not shown). Similar foreshock phenomenon was described by Fairfield et al. (1990). Apparently, the foreshock pressure pulses were further transported by the solar wind to the magnetosheath and could affect the magnetopause. Similar foreshock pressure pulses and their compression effects in the magnetosphere-ionosphere were reported by Korotova et al. (2011). "

Also, we revised correspondingly text around Figure 7 in section 2.5 (Line 366-370, and etc.):
"As one can see, most of the magnetic peaks at panel d and/or magnetosheath ions at panel b were preceded by the foreshock pressure pulses within 1-5 min (panel f), for example at ~1549, ~1611, ~1625 UT and etc. (see Table 2). There are exceptions for plasma penetrations #6 at 1648 UT and #7 at 1651:30 UT. Note that those events were preceded by IMF discontinuities as one can find in rotation of the cone angle (panel e) at 1645 and 1650 UT, respectively."
(Line 432-434):
"Most of the penetrating magnetosheath jets correspond to the foreshock pressure pulses. All jet-related plasma structures caused local compression effects at the dayside."

We add corresponding commentaries in Discussion section (Line 495-509):
"Comparative analysis of the THEMIS, OMNI and ACE data showed that the geomagnetic perturbations were not driven by the dynamic pressure of the pristine solar wind. Note that significant discrepancies between the OMNI data and THEMIS near-earth observations under quasi-radial IMF were reported frequently (e.g., McPherron et al., 2013; Suvorova and Dmitriev, 2016). THEMIS observations show firmly that geomagnetic perturbations were rather related to changes in the IMF cone angle and pressure pulses in the subsolar foreshock.
We demonstrated that in the magnetosheath, foreshock pressure pulses could be transformed to fast and dense magnetosheath streams, so-called jets. We found that 5 out of 7 magnetosheath jets were preceded by the foreshock pressure pulses. These results support well the previous findings that the plasma jets are typical consequence of the foreshock dynamics and variations in the IMF orientation (e.g., Fairfield et al., 1990; Lin et al., 1996; Archer et al., 2012; Dmitriev and Suvorova, 2012; 2015; Plaschke et al., 2018). In addition, similar effects of the foreshock pressure pulses and magnetosheath jets in the magnetosphere were reported (e.g., Sibeck and Korotova, 1996; Korotova et al., 2011; Heitala et al., 2012)."

Referee:
In addition, at L477 the authors state: "*we expect that variations in the geomagnetic field (if any) should result from the local magnetosheath pressure pulses.*" Why? There are a number of other mechanisms that can cause activity in ground magnetometer during relatively quiet times (e.g., ULF waves driven by Kelvin Helmholtz; ULF waves generated internally by plasma instabilities, etc…). Have the authors considered any of these? Why do they believe that these are not occurring at the time of the ground perturbations?

Reply
We have revised the statement and put it to the end of Section 2.6:
"Thus, the low and middle latitude ground-based magnetic observations demonstrate similarity in the magnetic variations of "dimple-hump" pattern at 1200-1600 UT with the satellite observations in the dayside magnetosphere. It should be noted that the magnetic peaks are not regular and are characterized by periodicities of tens of minutes that distinct them from magnetospheric quasi-periodic ULF waves with periods 1 – 600 s. Hence, the variations observed in the geomagnetic field should result from pressure pulses of foreshock and/or magnetosheath origin."

***Minor Comments by Referee3:***

- Figure 3 caption: "The shaded box denotes the time interval from 13 to 23 UT" – there is no shaded box in the figure

Reply:
The box was missed accidentally during the previous revision of the manuscript. We correct the Figure 3.

- L171: "quite" -> "quiet"

Reply:
Corrected.

- L181-182: You might mention here that this is the "0-degree telescope," since this is how it is commonly referred to in the literature.

Reply:
We change accordingly: "The data shown in Figure 1 are from the 0-degree telescope oriented along the orbital radius-vector…"

- L183: Is this the definition of the forbidden zone? If so, you should state that "The forbidden zone is defined as L<1.2 ….." What field model are you using to define the L values?

Reply:
We change the text accordingly:
"The forbidden zone is defined as $L < 1.2$ in the longitudinal range from 0° to 260°E (or 100°W) that is beyond the South Atlantic anomaly (SAA). The drift L-shells are calculated from IGRF-2005 model."

- L193-194: "*Fluxes of the >100 keV electrons and >30 keV protons did not increase also (not shown).*" You should indicate here whether you are referring to quasi-trapped, precipitating, or both.

Reply:
We clarify this accordingly: "Fluxes of the precipitating and quasi-trapped >100 keV electrons and >30 keV protons did not increase also (not shown)."

- L199: This labeling of the POES vehicles is not standard. Is this what you mean? "(P2 = MetOp2, P5 = NOAA-15, P6 = NOAA-16, P7 = NOAA-17, P8 = NOAA-18)"? If so, you should state this here.

Reply:
Corrected.

- L220-221: "All remaining enhancements F2, F3, F5, F6, F8 and F10 of >30 keV electron fluxeswere observed in the early morning (5 LT) for a long time interval of ~4 h" – I don't know how you can easily see this from the figure. I think you need to label each of the curves in Figure 2(a) with the corresponding FEE number.

Reply:
These labels were missed accidentally during the previous revision of the manuscript. We correct the Figure 2.

- L283: How do you know that TH-C is upstream of the bow shock? There's no bow shock model shown. Is this simply inferred from the TH-C measurements?

Reply:

Yes, it is inferred from the observations. It is also supported by the average location of the subsolar bow shock from the paper by D. Faifield (1971). We clarify this in the text around Figure 5 (Line 236-244):

"During the time interval from 1200 to 1800 UT, the THEMIS-C satellite (TH-C) moved from the subsolar region (17.2, -0.3, -5.9 Re GSM) toward dusk (18.1, 3.4, -5.9 Re GSM) (see Figure 4). From the TH-C plasma and magnetic measurements (Figure 5), we infer that the probe was located upstream of the bow shock, whose average subsolar position was estimated as ~14.6 Re for $P$d~1.5 nPa (Fairfield, 1971). Figure 5a shows measurements of the THEMIS-C/FGM fluxgate magnetometer in GSM coordinates with a time resolution of ~3 s (Auster et al., 2008) and the ion spectrograms from THEMIS-C/ESA plasma instrument (McFadden et al., 2008). The ion spectrogram clearly demonstrates that hot ions (~ 1 keV) are of the solar wind origin and magnitudes of magnetic field components correspond to IMF components in Figure 3."

- L287: "GMS" -> "GSM"

Reply:
Corrected.

- L300: "After 1500 UT, the OMNI data do not match the TH-C observation any more, even with time correction." – it would be nice if you showed also a smoothed version of the TH-C cone angle time series in Figure 5(c )

Reply:
We add the smoothed curve in Figure 5c.

- L401: Are the ACE data time shifted here?

Reply:
Yes. We add in the text (Line 286) and caption of Figure 5: "The ACE data are shifted by 60 min."

---

## Author Response (AR3)

**Reply to Reviewer #3**

We thank the reviewer for comments. We try to address all the Reviewer's comments and criticism.

*The observations presented in this paper that show the appearance of >30 keV electrons in the "forbidden" region during quiet times is very interesting and worthy of publications. I thank the authors for attempting to respond to my earlier criticisms, but I remained unconvinced of their claims relating the low L injections to dayside magnetosheath jet activity, which, through a complicated chain of events, is supposed to enhance the low latitude electric field on the nightside. I attempt to summarize my objections as follows.*

*As I see it, these facts are supported by the evidence presented manuscript, all of which I agree with:*
*• >30 keV electrons were observed at very low L, L<1.2 during a quiet interval*
*• They were likely injected from the nightside in the 2-5 MLT region*
*• They were not associated with substorm injection/activity*
*• There were global magnetic field perturbations observed throughout the dayside magnetosphere (GOES, THEMIS, ground mags) around the same time*
*• These global field perturbations were likely related to upstream foreshock activity/waves*
*• The field perturbations were too weak to produce radial transport of >30 keV electrons and were not the cause of the low L injections*
*• Foreshock pulses and associated magnetosheath jets were observed on the dayside*

*These represent very interesting and intriguing observations, particularly the appearance of electrons at very low L during quiet time.*
*However, it is then argued that the magnetosheath jets cause hot plasma (50 eV - 10 keV) to precipitate into the dayside auroral region (L = 7 - 15) and that the jet-related magnetosheath plasma can produce significant additional ionization and increase conductivity of the high-latitude ionosphere on the dayside. It is then argued that this enhanced dayside conductivity enhances dayside currents in the ionosphere which "should in turn promote generation of transient localized electric fields on the nightside and especially in the postmidnight sector, where the conductivity is weak." I do not follow this logic and there are no additional arguments/calculations/references to support these claims.*

Yes, we agree with the reviewer that the jet-related auroral precipitations observed by POES (~1 erg/(cm$^2$ s)) are not strong enough to induce a strong nightside electric field. We comment this important issue in the revised manuscript:

"We should point out that the scenario suffers some shortcomings. The energy flux of auroral precipitations of ~ 1 erg/(cm$^2$ s) was observed to be weak relative to that during substorms that results in a relatively weak additional ionization in the dayside ionosphere. It is hard to expect that the weak increase in the ionization can induce strong electric field of $E$ ~ 5mV/m. On the other hand, the satellite observations are sparse in space and time and, thus, a satellite might not catch an intense jet-related localized auroral precipitation of ~10 min duration. Hence, the experimental information about auroral precipitations on the dayside is still incomplete. "

*It is then hypothesized that "the induced nightside electric field might penetrate from high to low latitudes (very low L shells) and results in ExB drift of electrons to lower L-shells."*
*I do not understand the mechanism that would allow this localized nightside electric field to penetrate from high to low latitudes. Again, there are no additional arguments/ calculations /references to support these claims. It is then argued that it is this electric field that produces the electron injections at very low L.*

The origin of strong electric field at L < 1.2 is still totally unresolved problem. However, the existence of this electric field is already accepted and widely used by the scientific society (e.g. Selesnick et al., 2019). Apparently the resolving of this problem is beyond the scope of our study. We can only make some assumptions. This important issue is discussed in the revised manuscript:

*"Another serious problem is the generation/penetration of electric fields in the inner*
magnetosphere at low latitudes in the night sector, which is far from complete understanding.
The convection electric field of up to 2 mV/m was observed at $L > 2$ during disturbed
geomagnetic conditions (Califf et al., 2014; 2017). During magnetic quiet, the convection
electric field is apparently smaller (<0.5 mV/m). On the other hand, prompt penetrating electric
field in the dayside ionosphere at heights ~100 km was estimated of ~2 mV/m (Huang, 2008).
However, electric field at heights from 1000 to 2000 km did not measured and, thus, its value is
unknown. There are also no models predicting strong electric fields in the inner radiation belt
and below. As conjugate observations of penetrating transient electric fields are still unavailable
for such cases of anomalous particle transport, the exact mechanism of deep electron injections
cannot as yet be fully determined.*"*
*In summary, I find these final arguments regarding the last chain in the (complicated) proposed scenario*
*to be weak and unconvincing.*
We figure out this point in the end of the paper:
"Summarizing, from the experimental data available, the existing scenario cannot be supported
firmly. It might also be that another unknown mechanism is responsible for the FEE
enhancements during magnetic quiet periods. In this sense, further experimental studies and *in*
*situ* observations of electric fields at *L*-shells from 1.1 to 2 as well as of dayside auroral
precipitations are required."
We also discussed these issues in Introduction (e.g., Lines 111 – 123) and Discussion.
We should emphasize that we consider a qualitative scenario, which is based on our previous
publications. Ground magnetic and radar observations showed that electric fields penetrate from
high to low latitudes (e.g., Huang, 2008). It should be addressed that mechanisms are currently
under comprehensive investigations. For example, possible mechanisms of penetration of
electric fields can be found in the review paper by Kikuchi and Hashimoto (2016). Recently, a
new mechanism of electric field penetration during northward IMF was suggested by Huang
(2019). Probably, a specific mechanism is needed for this particular case, but this can be a
subject for future studies in the case if the manuscript will be available for scientific discussions.
In our interpretation of the observations, we follow the logic that electric field and conductivity
are interconnected phenomena. Yet it relies on findings published in our previous papers
(Suvorova et al., 2016; Suvorova 2017) and other studies (e.g., Sibeck et al., 1996; Vorobjev et
al., 2001; Han et al., 2018; Selesnick et al., 2016, and etc.) as cited in the text. We would like to
hope that additional arguments or contra-arguments with model calculations will appear in future
studies, because the observations present challenges for current models of electric field and
electron injections below L<2 under quiet solar wind conditions.
In the revised manuscript, we explain:
"It should be noted that most favorable conditions for FEE enhancements (and, presumably, for
penetration of localized electric fields) arise in the period from May to September independently
on geomagnetic activity level (Suvorova, 2017) Similar asymmetry in the dayside auroral
conductivity was also shown by Sibeck et al., (1996). Our case event on 1 August 2008
corresponds well to these favorable conditions. Taking into account our previous finding that the
occurrence of FEE enhancements is related to the ionization of the dayside ionosphere at high
latitudes (e.g. Suvorova, 2017), the following scenario can be considered:"

*I will also comment that I see no relationship in Figure 11 between the magnetic field perturbations and*
*the NOAA/POES/TED precipitation signatures.*
*It is not demonstrated whether these TED precipitation signatures are exceptional or the norm. What do*
*the TED measurements show before and after this interval? When the magnetic field is quiet, are these*
*plasma precipitations observed? I suspect that the TED measurements always look like this, but one*
*cannot be sure from the manuscript. If they are, then that begs the question why does the mechanism*
*proposed by the authors only occur in this event, and not all of the time?*
*What is so unique about the magnetospheric state and the observations that allow access of >30 keV*
*electrons down to very low L values in this rare event?*
*In summary, while I think that the appearance of electrons at very low L during quiet time is a very*
*interesting question, the authors have not convinced me that their proposed scenario is plausible, and*
*thus I cannot recommend this article for publication.*

The main intrigue of this event is that this particular interval was accompanied by foreshock
pressure pulses and by magnetosheath plasma jets. They did not occur on August 2. The hot
plasma precipitations measured by the POES/TED instrument on August 1 and 2 are shown in
Figure S3 (see supplement and Figure 1 below). Dayside high-latitude precipitations are marked
by white circle. One can see more intense precipitations with energy flux of >0.5-1 erg/cm$^2$ sr at
latitude around 76º (dashed line) and at longitude ~40ºW and ~10ºW during 1 August (against
energy flux of <0.1 erg/cm$^2$ sr during 2 August).

[Figure]

**Figure 1.** Global maps of energy flux of hot plasma precipitations obtained from NOAA/POES
satellites from 12 to 16 UT on 1 and 2 August 2008 (left and right, respectively). More intense
precipitations with energy flux of >0.5-1 erg/cm$^2$ sr were observed at latitude around 76º (dashed
line) and at longitudes ~40ºW and ~10ºW during 1 August (against energy flux of <0.1 erg/cm$^2$
sr during the 2 August).

In order to demonstrate the unique magnetospheric state, we also show precipitations of >30 keV
protons and electrons obtained from NOAA/POES satellites during the interval from 12 to 16
UT on 1 and 2 August (see Figures S4 and S5 in supplement and Figures 2 and 3 below). In
Figure 2, localized proton precipitations near noon are found at L~6-8 at 1330 UT on August 1,
while they are absent on August 2.

[Figure]

**Figure 2.** Proton fluxes with energy 30-80 keV obtained from (a) NOAA/POES-18 and (b) METOP-02 in the dayside sector on 1 and 2 August 2008. Trapped (precipitating) protons are shown by thin (thick) curves. The time moments at 13 LT and 11 LT correspond to enhanced precipitation at L~8 on August 1.

Proton and electron precipitations are shown on the global maps in Figure 3. In Figure 3a, one can see the region of intense proton fluxes around 60º of latitude (L ~ 6) within the range of 20º-80ºW of longitude (near noon) on August 1, which is detached from the higher latitude "isotropic" proton fluxes associated with the plasmasheet. Such detached precipitations are absent on 2 August. In other local time sectors both on August 1 and 2, precipitations occurred only in the high L-shell region (L >14), i.e. from the plasmasheet. In Figure 3b, global maps of >30 keV electron fluxes during the same interval are shown. It seems there are no features in electron precipitations on August 1 in comparison to August 2, may be except of a spot at ~20ºW of longitude (marked by white arrow).

**Thus, there is a notable difference in the magnetospheric state during two days: in location and enhanced flux of the energetic particles and hot plasma precipitations in the high-latitude region on the dayside.**

[Figure]

**Figure 3.** Global maps of (a) proton fluxes with energy 30-80 keV and (b) >30 keV electron fluxes obtained from NOAA/POES satellites from 12 to 16 UT on 1 and 2 August 2008 (left and right, respectively). Proton and electron precipitations (marked by white arrows) were observed on the dayside on 1 August. There were no prominent precipitations near noon on 2 August.

However, electric field at heights from 1000 to 2000 km did not measured and, thus, its value is
unknown. There are also no models predicting strong electric fields in the inner radiation belt
and below. As conjugate observations of penetrating transient electric fields are still unavailable
for such cases of anomalous particle transport, the exact mechanism of deep electron injections
cannot as yet be fully determined."
Concerning to recommendation to use other events, we should explain the following:
This study is devoted to a unique case event of long-lasting energetic electron enhancements
under the IRB during very quiet geomagnetic conditions. Actually, this interval includes 8
independent cases of energetic electron injections under various geomagnetic and upstream
conditions. There are no other events of such kind when we can use THEMIS data successfully.
*If a strong electric field exists, other energy ranges of electrons and ions should also be*
*accelerated. Also, flux enhancements should occur at all L-shells. However, there is no evidence*
*of flux enhancements in other energies, species or L-shells. The paper needs to provide a*
*mechanism of how flux enhancements can occur without affecting other energies of electrons or*
*ions. An investigation of NOAA fluxes at higher L-shells are also needed to check if the slot*
*region and the outer radiation belt responded. The authors mentioned that the ring current flux*
*enhancements (and thus ENA flux enhancements) aren't expected. It is hard to understand why*
*the inner belt can respond to pressure pulses without affecting the ring current.*
Electrons with higher energies have a much shorter period of the azimuthal drift that makes
difficult for them to stay in the localized region of abnormal radial transport. We discussed it in
Lines 574-579:
"The multi-step process is limited by the time, during which a particle stays in the region of
injection. The >30 keV electrons have a long period of azimuthal drift and, thus, they can stay in
the region for hours. In contrast, the >100 keV electrons with the azimuthal period of ~6 h leave
quickly the injection region and, thus, do not have enough time to penetrate to the forbidden
zone. This effect can explain the absence of high-energy electrons in the FEE enhancements
presented."
Concerning to the extension in L-shells ("*An investigation of NOAA fluxes at higher L-shells are*
*also needed to check if the slot region and the outer radiation belt responded."*).
We clarify this important issue in the text:
"In the case of electric field penetrating from high to lower latitudes, the following effect might
be important. At higher altitudes (larger L-shells), the azimuthal drift periods of particles
decrease dramatically. Hence, the particles escape quickly from the localized region with the
enhanced electric field and, as a result, they drift earthward only a little."
The problem of protons is discussed in Introduction:
"From a comparison of deep penetrations of electrons and protons, Zhao et al. (2017a) have
revealed principle differences in these phenomena suggesting different underlying physical
mechanisms responsible for deep penetrations of protons and electrons. Particularly, deep proton penetration is consistent with convection of plasma sheet protons, and deep electron penetration
suggests the existence of a local time localized mechanism."

Concerning to the ring current problem ("*The authors mentioned that the ring current flux*
*enhancements (and thus ENA flux enhancements) aren't expected. It is hard to understand why*
*the inner belt can respond to pressure pulses without affecting the ring current.*")
In the manuscript, we did not mention the "*ring current*" because there was no any ring current
at that very quiet day (see a map in Figure below). What line # of the manuscript does the
reviewer mean?

[Figure]

**Figure:** Global map of >30 keV proton fluxes measured by the detector-0 on board
NOAA/POES satellites on August 1, 2008 from 12 to 24 UT.

*The correlation between pressure pulses and flux enhancements is interesting if it is real, but it is*
*difficult to draw a firm conclusion from the limited event presented in this paper. The time*
*interval of interest includes PC index enhancements due to the southward IMF. The convection*
*electric field can increase under southward IMF, though it won't be as large as 5 mV/m. The*
*authors didn't rule out the possibilities that non-pressure-pulse effects are responsible for the*
*flux enhancements. It would be necessary to analyze more events to clearly show that pressure*
*pulses are the only cause of the flux enhancements.*

As we mention above, this event is unique. There were 8 independent injections during one quiet
interval. We should note that availability of observations near the dayside bow shock is crucial in
this type of events and such opportunity does not always exist. Fortunately, THEMIS-C was
located in the right place in the event presented.
Statistical investigation of several events will be a subject of further study. We cannot put
everything in one paper because it will be enormously large.

Concerning to southward IMF. Indeed, according to OMNI database (Figure 3c and 5b) IMF Bz
changed a sign after 1420 UT. However, a notable sharp increase of the PC index occurred at
~1400 UT (Figure 3d) and, moreover, the first FEE enhancement occurred at ~1245 (Figure 2).
According to the THEMIS-C observations near the bow shock (Figure 5a), the southward
turning occurred even later, at 1550 UT. In Lines 280-284 we noted that IMF Bz was positive at
least until 1440 UT. The THEMIS observations convincingly prove that the PC index and FEE
flux enhancements during 1300-1600 UT were by no means related to southward Bz effect. On
the other hand, the dayside magnetospheric magnetic field pulses evidenced certainly the
pressure-pulse effects (Figure 6). Only based on these observations, we rule out the southward
IMF as a possible "non-pressure-pulse" reason for the flux enhancements in the interval 1300-
1600 UT.

*Line 479: If pressure pulses cause the ground magnetic field perturbation, the largest magnetic*
*field signal should occur near noon, but the actual largest signal was measured near dawn. It*
*doesn't support the pressure pulse source but some other phenomena are more important for*
*creating the ground magnetic field changes.*
This is actually misinterpretation of Figure 9. It is originated from the different scaling for
different stations. Moreover, we did not state in the paper that "*actual largest signal was*
*measured near dawn*" (see Lines 479 – 480).
To clarify better this point, we present magnetic variations at three INTERMAGNET stations
during a shorter 2-h interval in Figure below. It shows the H-component of the geomagnetic field
at KOU, ASC and PPT from 1300 to 1500 UT. One can see that at the dayside stations KOU
(LT=9.5-11.5 h) and ASC (LT=12-14h) the peak-to peak amplitude (~3-4 nT) is larger than the
amplitude (~2 nT) at the morning station PPT (LT=3-5 h).

[Figure]

**Figure:** A version of Figure 9 for three INTERMAGNET stations during 2-h interval from 13 to
15 UT. The H components of geomagnetic field at stations ASC, KOU, and PPT are shown.
*High-latitude magnetometer data should also be presented. Although the authors state that*
*substorms occur after the flux enhancement, in the equivalent current maps in the SuperMAG*
*website, the largest enhancements at ~13:30 UT were seen in the nightside, while angle changes*
*were seen in the dayside and dawnside. This plot suggests that a substorm-like nightside high-*
*latitude auroral activity was present. The authors should discuss how it may be related to flux*
*enhancements. The manuscript repeatedly mention injection, but there is no discussion about*
*how injection and pressure pulses are causally related. The analysis doesn't rule out the*
*possibility that injection is not related to pressure pulses but is caused by independent nightside*
*processes.*
We very appreciate the Reviewer for recommendation to use the SuperMag website.
We compare geomagnetic activity on 1 August with that on 2 and 3 August after 1200 UT.
Corresponding geomagnetic SME indices are shown in Figure below. Geomagnetic activity in
the interval 12-20 UT is similar on 1 and 3 August. The quietest day is 2 August.

[Figure]

We have considered the magnetic data at high latitudes provided by SuperMAG website at various time intervals during 1, 2, and 3 August 2008 (see Figure below and Figure S1 in Supplement). In Figure below, we show time moments for FEE enhancements at 1330 UT (upper row) and for substorm-like event at 1700 UT (lower row). It can be clearly see that in contrast to substorm-like event (1700 UT), the high- and mid-latitude magnetic activity is weak during FEEs (1330 UT) on 1 and 3 August. This activity is comparable with the very quiet period on 2 August.

[Figure]

In Figure S1 (see Supplement), we show time moments at 1305, 1330, 1430 and 1540 UT during 3 days. Only one general feature can be pointed out during FEE enhancements during 13-16 UT: prominent magnetic activity in polar region at noon and in the dawn sector. This activity is definitely not related to substorms in the magnetotail but rather to the compression of the dayside magnetosphere.

Also, one can found that the largest enhancements in the nightside sector at all selected times during different days (even on August 2) were systematically seen at the same single station (in Alaska region). We believe that this unlikely relates to substorm-like activity. This artificial effect possibly relates to incorrect treatment of the background level at this particular station.

Concerning to substorm, we note the following in the beginning of Discussion section:
"It is important to note that the intensification of AE index from 1600 to 1800 UT was originated from magnetic activity at high latitudes on the dayside (see Figure S2 in Supplement). The dayside activity results from the multiple magnetospheric compressions (see Figure 6). In this context, the substorm should be rather considered as a "substorm-like" event related to compressions of the dayside magnetosphere."

*The peak magnetic field perturbation occurs near dawn and the authors inferred that the injection occurred at 0-6 LT. But if upstream pressure pulses drive injection, the peak magnetic field perturbation should be seen near noon. The authors should discuss how pressure pulses cause nightside injection without much dayside perturbations. Nightside auroral activity could occur without a causal connection to pressure pulses. This possibility should be discussed.*

We clarify this issue above (see the reply to comment "Line 479"). The dayside perturbations are larger than nightside perturbations (see Figure in the reply). The activity was related to the compression rather than to night-time substorm activity.

*Line 1 under radiation belt -> earthward of the inner radiation belt*

We thank the referee for the suggestion. Another alternative variant was used in the paper by Selesnik et al. (JGR, 2019) "Energetic Electrons Below the Inner Radiation Belt". One can often find "under the inner radiation belt" or "below the inner radiation belt" in literature. We think that both variants are appropriate.

*Line 2 nonstorm -> a nonstorm*
Corrected

H, nT

B, devB, nT

TH-D

G12

11h    12h    13h    14h    15h    16h

**Figure 9.** Relative variations in the horizontal component (H) of the geomagnetic field at low geomagnetic latitudes. Local time intervals are indicated near the station codes. The vertical lines depict magnetic peaks #1 - #3 at THEMIS (see Table 2). Bottom panel shows magnetic field B measured by GOES-12 (black) and detrended magnetic field from TH-D (green).

[Figure]

**Figure 10.** Relative variations in the horizontal component (H) of the geomagnetic field in the midnight (left) and predawn (right) sectors. The geomagnetic latitudes of the stations are indicated near station codes. The vertical lines depict magnetic peaks at THEMIS (see Table 2). Magnetic data from THEMIS and GOES satellites are shown at lower panels on the right.

[Figure]

**Figure 11.** Dynamics of the geomagnetic field and particles on 1 August 2008: (a) FEE enhancements, (b) plasma precipitation at high latitudes, and dayside magnetic field perturbations observed by (c) GOES-12 (black), TH-D (green) and TH-B (brown). The left y-axis corresponds to GOES-12, and the right y-axis to TH-D and TH-B. The numbers indicate the FEE injections at ~2 and ~5 LT (see Table 1), colors for POES satellite are the same as in Figure 2. Plasma precipitations are shown for the energy flux above the threshold of 0.5 (erg/sm$^2$ s) and are grouped in LT: 23 – 24 LT (light gray), 0 – 2 LT (gray), 5 – 6 LT (blue), 12.5 - 15 LT (red points), 15 – 16 LT (violet), and 19.5 – 21.5 LT (green).

---

## Author Response (AR4)

**Reply to reviewer #4**

We thank the reviewer for the comments. We try to address all of them and clarify the relevant issues.

*The authors made some clarifications and I mostly agree with each piece of observation. However, the scenario of magnetosheath jets accelerating >30 keV electrons at low L is still not supported by concrete evidence of the direct causal relation. (1) The authors admit that there are no electric field data, no other days of events despite a frequent occurrence of jets, and no acceleration at other locations or energies. (2) Other possible scenarios aren't firmly ruled out because most of the FEEs occurred during the PC index increase and substorms. Currently the proposed scenario is a speculation at best. I suggest to weaken the abstract and conclusion. Those sections should clearly state that (1) there is no direct evidence that supports electron acceleration by magnetosheath jets and (2) PC index increase and substorms could also contribute to acceleration.*
*...*
*The PC index increased at 13 UT and substorms started at 16 UT. Except for the 1245 UT FEE, all other FEEs occurred during the PC index increase or substorms. As in the SuperMAG plots, the high-latitude magnetic field increased at all MLT (indicating convection enhancements) or more pronounced near midnight (substorms). How do the authors rule out the possibilities that the FEEs after 13 UT are affected by the PC index and substorms?*

We agree with the reviewer that the proposed scenario, in particular no.4 and no.5 (Lines 607-612), "*is not supported by concrete evidence of the direct causal relation*". Note that we fully expressed this in the final conclusion of the manuscript (Lines 631-635):
"Summarizing, from the experimental data available, the existing scenario cannot be supported firmly. It might also be that another unknown mechanism is responsible for the FEE enhancements during magnetic quiet periods. In this sense, further experimental studies and *in situ* observations of electric fields at *L*-shells from 1.1 to 2 as well as of dayside auroral precipitations are required."
In the abstract of the revised manuscript, we add:
"However, the scenario cannot be firmly supported because of the lack of experimental data on electric fields at the heights of electron injections. This should be a subject of future experiments."

However, we cannot agree with the reviewer's statements that
"*most of the FEEs occurred during the PC index increase and substorms*"
and "*PC index increase and substorms could also contribute to acceleration.*"
The first injections of >30 keV electrons at 1250 and 1315 UT (see Table 1) were not preceded or accompanied by any notable activity in PC and AE indices (see Figure 3). Hence, those injections were definitely not related to any disturbances at polar and auroral latitudes. Other FEEs from Table 1 were accompanied by enhancements in PC index but they were not related to a substorm because the increase of AE index from 1600 UT to ~ 1730 UT was originated from the dayside stations and, thus, it was definitely not a substorm.

We expected that these important issues had been addressed clearly enough both in the manuscript and in our reply to the comments from the previous review. It seems, the Reviewer did not completely understand our explanations or, perhaps, did not agree with them, despite of a lot of graphic materials putted in the Supplement, where the magnetometer data from SuperMag were presented in Figures S1 and S2. Below, we reply to the relevant comments with more details.

**I  There was no substorm activity from 1200 UT to ~1730 UT**

In the previous reply, we pointed out a systematic error originated from a single station, which permanently showed a disturbance of ~100 nT during successive days. That station is T41 (Kiana) locating at the low-latitude edge of the auroral oval (glon=199.6º, glat=67º, mlon=-105.55º, mlat=65.62º). Let's look at Figure 1R below, which shows examples of magnetic data plots for different time moments on 1 August 2008. The T41 (marked by red asterisk) was situated in different local time sectors. When the station is at night side (for example, at 10 UT and after), it is easy to mistake the T41's signal for a substorm signature. However, the T41's magnetic vector did not change from 0 UT (local noon) to 16 UT (local dawn), while neighboring stations showed random temporal variations rather then typical substorm dynamics. The substorm onset never occurs at dayside, and hence there is no substorm signature at all during the whole interval. The same feature at other UT times and days was shown in the previous reply. It seems that the T41 magnetic data were erroneously represented.

[Figure]

**Figure 1R**. Magnetic data vectors provided by SuperMag at high geomagnetic latitudes (>60º). Plots are shown for different times (0 UT, 4, 10, 12, 12:30, 13, 14, 15 and 16 UT) on 1 August. Red asterisk indicates T41 (Kiana) station. Three stations inside the blue contour are BRW, KAV and INK stations.

In order to understand the reason of this strange feature, we check magnetic records of this station and other three stations BRW (Barrow), KAV (Kaktovik) and INV (Inuvik), which are located at the polar edge of the auroral oval at mag. latitude around 71º (they are inside the blue oval in Figure 1R). In Figure 2R, the higher latitude stations recorded a negative bay in the north-south component after 16 UT in the dawn sector (between 3.5 and 5 LT at 16 UT). While the T41 did not see any prominent disturbance in the auroral oval. Hence, we can conclude the following:
1. The vector representation of T41 in polar coordinates (Figure 1R) is wrong because it contradicts to the time profiles (Figure 2R).
2. If T41 is eliminated from the consideration then the most prominent magnetic activity occurs on the dayside from 14 UT and later by 1730 UT, the activity spread to the nightside through the dawn and dusk sectors (see Figure S2 in Supplementary materials). This dynamics does not definitely proper for substorm.

[Figure]

Figure 2R. The magnetometer data for BRW(70.5º), KAV(71.4º), INK(71.4º) and T41(65.5º) stations.

**II  The increase of PC index results from magnetospheric compressions**
The PC index is a measure of the energy inflow from the solar wind into the Earth's magnetosphere and describes magnetic disturbances due to ionospheric and field-aligned currents at the polar cap. The PC and AE indices are a measure of geomagnetic disturbances in different regions (polar cap and auroral zone), so the PC index does not represent substorms in the same way as the AE index.

In order to clarify this very important issue, we add the following paragraph around Figure 3: "As shown in Figure 3, the polar cap PCN index started to increase after 1300 UT under northward IMF. After 1400 UT, the moderate polar cap activity (PCN~1.5-2 mV/m) indicates intensification of the R1 field-aligned currents in the dawn and dusk magnetosphere (Troshichev et al., 2016). It should be noted that the weak and moderate PC-index activity can be also produced by changes in the solar wind dynamic pressure (Lukianova, 2003). Hence, the enhanced PCN during 1300 - 1600 UT might indicate the compressions of dayside magnetosphere. However, from Figure 3, it is difficult to identify appropriate solar wind drivers for interpretation of the polar cap activity at that time. From analysis of SuperMag magnetic data, we found that the magnetic variations dominated on the dayside, dawn and partially dusk sectors from 1300 to 1700 UT (see Figures S1 and S2 in supplementary material). Hence, the enhancement of PCN index from 1300 to 1600 UT resulted rather from compressions of the dayside magnetosphere."

Hence, we can rule out the PC increase as a driver because in the present case, the PC increase is a manifestation of the magnetospheric compression.

**III  The substorm-like event was originated from dayside magnetosphere compressions**

The auroral activity in AE index after 16 UT resembled an isolated substorm. However, analysis of the SuperMag data (see Figure S2 in Supplementary materials) shows that the magnetic activity occur mainly on the dayside. In the manuscript, we wrote in the second paragraph of Discussion section:

"It is important to note that the intensification of AE index from 1600 to 1800 UT was originated from magnetic activity at high latitudes on the dayside (see Figure S2 in Supplement). The dayside activity results from the multiple magnetospheric compressions (see Figure 6). In this context, the substorm should be rather considered as a "substorm-like" event related to compressions of the dayside magnetosphere."

Actually, this was our reply to the comments from the previous review. In the present review, the Reviewer ignores the arguments presented in our reply and again arises this problem.

We will very appreciate the Reviewer for indication of any mistakes in our argumentation.

Concerning "*no acceleration at other locations or energies*".

We wrote in Introduction that other authors concluded that the injection mechanism operating at low L shells is totally different from that operating at high L shells, and energy range is also different. It does not depend on substorm occurrence!

It is very hard to imaging that energetic particles can be transported from the plasmasheet at $L \sim$ 10 to the inner range at $L \sim 1.1$ within 30 minutes. Actually, it is impossible because the $ExB$ drift speed is limited. This is why we should not need to consider the dynamics of the outer radiation belt. This important issue as well as other relevant effects has been considered in the Discussion of original manuscript. However, the Reviewer totally ignored our arguments. Unfortunately, the Reviewer insists to use this wrong idea for criticism of our results.

Concerning *"no other days of events despite a frequent occurrence of jets"*, we wrote in Introduction that the external drivers (solar wind / foreshock pressure pulses and jets) are only necessary but not sufficient for FEE injections.

In order to conduct the similar study one should find a time interval, which is satisfied to the following requirements:

1. FEE enhancement
2. THEMIS should be in right place (dayside magnetosheath during Northern summer)
3. Conditions for jet generation should be satisfied
4. Quiet solar wind conditions with quasi-state dynamic pressure and northward IMF

5. Quiet geomagnetic conditions: no storms and/or substorms.

Apparently, it is extremely difficult to satisfy all this conditions simultaneously. For instance, there are only 1 or 2 geomagnetic quiet days per month.

This is why the statistical study is almost impossible. Among a few case events found in 2007 - 2009, this one is most representative. Namely, several FEE enhancements were observed during geomagnetic quiet. Other cases were accompanied by a single FEE enhancement.

The key suggestion in our scenario is about the electric field of a few mV/m at L<1.5. It is important evidence that we really don't have (it is unlikely that anyone will ever have this evidence). Experimental observation of electric fields at low heights could never be conducted in the past and it would hardly be conducted in the nearest future.

The statements "*the scenario of magnetosheath jets accelerating >30 keV electrons at low L*" is incorrect. We never use this formulation in the manuscript. Such simplified view essentially misstates our ideas about deep electron injections during the transient foreshock.

*The authors mentioned that the first FEE was at 1245 UT. But the paper doesn't show a magnetosheath jet at that time. The dynamic pressure at TH-C was constant, and the magnetic field increase at GOES was after 1255 UT. How do the authors explain that the 1245 UT FEE is caused by a magnetosheath jet?*

First of all, the statement "*and the magnetic field increase at GOES was after 1255 UT*" is rough distortion of the data presented. In the original manuscript, we clearly indicated magnetospheric compression and precipitations:

"From 11 to 13 UT, one can see several increases of a few nT observed by GOES and/or THEMIS at ~1125, ~1200, ~1245 and ~1300 UT (Figure 6b)."

"in Figure 11, we find two cases of geomagnetic pulses followed by intense dayside precipitations of the hot plasma at 1105 UT and 1145 UT."

Hence, the first FEE enhancements were preceded by the dayside magnetospheric compressions and high-latitude precipitations. It is very important to point out here that the response of FEE to the solar wind drivers is not instant. In Discussion, we estimate that "the earthward drift of energetic electron across the magnetic field lines from $L = 1.2$ to $L = 1.1$ takes up to 40 min". Hence, if the FEE is observed at 1250 UT (see Table 1) then the impact (compression and precipitations) should occur earlier.

The fact that TH-C as well as ACE did not detect any enhancements of the dynamic pressure in the solar wind from 1100 to 1320 UT (both right upstream of the subsolar bow shock and far upstream) has very important meaning: the driver of magnetospheric compression was not located in the solar wind. Then what is origin of the compressions and precipitations? Up to now, we know only two possible phenomena: 1). Pressure pulses in the subsolar foreshock and 2). Magnetosheath plasma jets. From 11 to ~1320 UT, there was no subsolar foreshock (see Figure 5). Hence, only magnetosheath plasma jets could cause the compression of the dayside magnetosphere.

[revised manuscript text omitted]